# LoRA vs Full Fine-tuning:
# An Illusion of Equivalence

## Abstract

Fine-tuning is a crucial paradigm for adapting pre-trained large language models to downstream tasks. Recently, methods like Low-Rank Adaptation (LoRA) have been shown to match the performance of fully fine-tuned models on various tasks with an extreme reduction in the number of trainable parameters. Even in settings where both methods learn similarly accurate models, *are their learned solutions really equivalent?* We study how different fine-tuning methods change pre-trained models by analyzing the model's weight matrices through the lens of their spectral properties. We find that full fine-tuning and LoRA yield weight matrices whose singular value decompositions exhibit very different structure; moreover, the fine-tuned models themselves show distinct generalization behaviors when tested outside the adaptation task's distribution. More specifically, we first show that the weight matrices trained with LoRA have new, high-ranking singular vectors, which we call *intruder dimensions*. Intruder dimensions do not appear during full fine-tuning. Second, we show that LoRA models with intruder dimensions, despite achieving similar performance to full fine-tuning on the target task, become worse models of the pre-training distribution and adapt less robustly to multiple tasks sequentially. Higher-rank, rank-stabilized LoRA models closely mirror full fine-tuning, even when performing on par with lower-rank LoRA models on the same tasks. These results suggest that models updated with LoRA and full fine-tuning access different parts of parameter space, even when they perform equally on the fine-tuned distribution. We conclude by examining why intruder dimensions appear in LoRA fine-tuned models, why they are undesirable, and how their effects can be minimized.

## 1 Introduction

Adapting large, pre-trained models to downstream tasks via fine-tuning is a computation- and data-efficient way to create domain-specific models for a variety of tasks. The simplest approach is to fine-tune all parameters of the pre-trained model on downstream task data (Devlin et al., 2019; Ouyang et al., 2022). However, as pre-trained models grow larger, full fine-tuning becomes increasingly challenging and expensive. Recently, parameter-efficient fine-tuning (PEFT) methods, especially low-rank adaptation (LoRA; Hu et al., 2021), have been shown to enable fine-tuning with only a fraction of the trainable parameters. **But even when fine-tuning with LoRA matches the performance of full fine-tuning, are the solutions learned by the two methods really equivalent?**

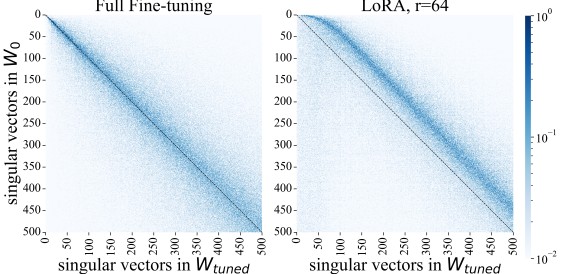

Figure 1: **Spectral dissimilarities between full fine-tuning and LoRA.** Similarity matrix of pre- and post-fine-tuning singular vectors of the weight matrices to characterize spectral differences introduced upon fine-tuning, in a representative example for LLaMA-2 fine-tuned on Magicoder. Full fine-tuning retains most of the pre-training structure; the diagonal shift in LoRA corresponds to the introduction of intruder dimensions. Color shows cosine similarity.

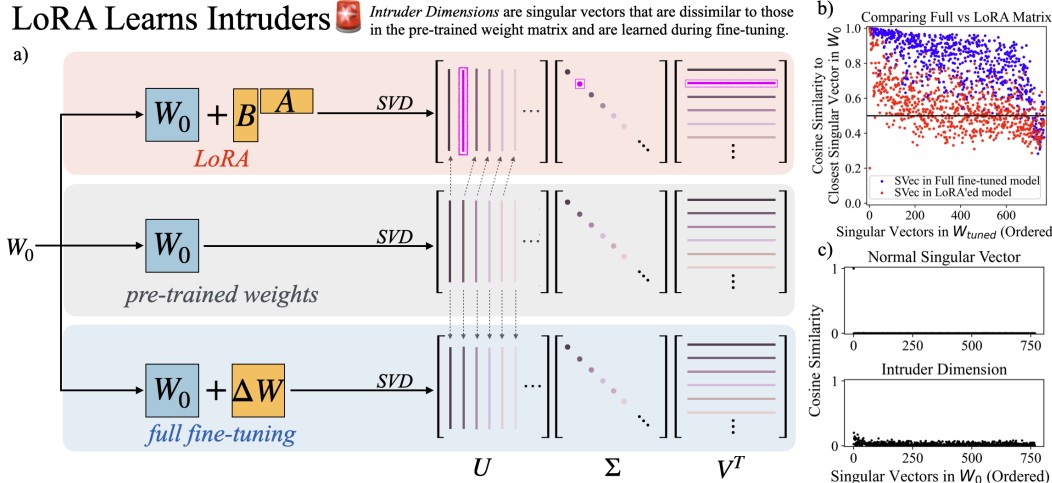

Figure 2: **Characterizing structural differences between solutions learnt by LoRA Vs full Fine-tuning. a)** We measure the changes to the SVD of the pre-trained weights made during fine-tuning. We observe *intruder dimensions* introduced by LoRA in top ranking singular vectors but by full fine-tuning. **b)** Comparing a matrix fine-tuned with full fine-tuning or LoRA. **c)** Comparing a normal singular vs an intruder dimension to all pre-trained singular vectors.

While full fine-tuning treats every parameter as trainable, LoRA treats the learned update to a weight matrix as the product of two low-rank matrices (Hu et al., 2021; Dettmers et al., 2023). While this parameterization is empirically effective, a principled explanation of the mechanism by which it matches the full fine-tuning performance has remained elusive. One explanation is offered by the *intrinsic dimension hypothesis* (Li et al., 2018; Aghajanyan et al., 2021), which posits that the update learned via fine-tuning has an intrinsically low intrinsic rank, suggesting that LoRA might recover an approximately equivalent solution to full fine-tuning. However, prior work has observed differences in the ability of LoRA and full fine-tuning to independently change the angle and magnitude with which a neuron transforms its input (Liu et al., 2024). Moreover, other work has also observed that LoRA has difficulty matching the performance of full fine-tuning on harder tasks, like code generation (Biderman et al., 2024; Zhuo et al., 2024) and long-form text generation (Ivison et al., 2023). Therefore, it is unclear if these findings indicate a limit in LoRA's ability to fit to a specific downstream task, or if these methods learn inherently different solutions.

In this paper, we show that full fine-tuning and LoRA learn different solutions with characteristic differences in their spectral properties (as shown in Fig. 1) and different generalization behaviors outside the target task distribution. We observe:

1. **LoRA and full fine-tuning produce structurally different parameter updates, characterized by the existence of *intruder dimensions***. These are singular vectors, with large associated singular values, that are approximately orthogonal to the singular vectors in a pre-trained weight matrix. In contrast, fully fine-tuned models remain spectrally similar to the pre-trained model and do not contain intruder dimensions.

2. **Behaviorally, LoRA fine-tuned models with intruder dimensions forget more of the pre-training distribution and exhibit less robust continual learning compared to full fine-tuning:** LoRA fine-tuned models with intruder dimensions are inferior to fully fine-tuned models outside the adaptation task's distribution, despite matching accuracy in distribution. However, higher-rank LoRA fine-tuned models, with identical adaptation task performance, more closely resemble fully fine-tuned models on these measures. Very high rank LoRA models, for e.g., full-rank LoRA, too forget more of their pre-training distribution—highlighting the fact that LoRA is not exempt from the general tradeoff between expressive power and generalization.

3. **Even when a low-rank LoRA performs well on a target task, a higher-rank parameterization may still be preferable.** While we observe that our low-rank LoRAs ($r \leq 8$) fit our downstream task distribution as well as full fine-tuning and high-rank LoRAs, using a high-rank ($r = 64$) leads

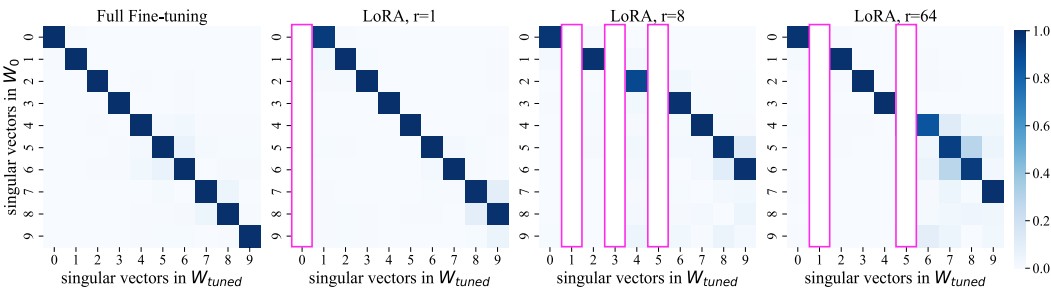

Figure 3: **Cosine similarities between sorted singular vectors in the fine-tuned models to pre-trained models.** *(Right)* Matrices fine-tuned with LoRA have a shift in singular vectors, as shown by blank columns, due to intruder dimensions (which are dissimilar to the pre-trained singular vectors). *(Left)* However, no such shift is found in the case of models trained via full fine-tuning.

to models that both exhibit better generalization and robust adaptability. However, in order to take advantage of higher ranks, the LoRA updated models must be rank-stabilized (Kalajdzievski, 2023).

## 2  BACKGROUND & RELATED WORK

**Methods for fine-tuning.** Pre-trained language models offer a foundation for downstream applications, eliminating the need to train from scratch (Ouyang et al., 2022; Devlin et al., 2019). Full fine-tuning, in which every parameter of a pre-trained model is updated, has been used for adaptation (Devlin et al., 2019; Liu et al., 2019). Low Rank Adaptation (LoRA; Hu et al., 2021), which represents the update to the weights as a product of two low-rank matrices, reduces computation and memory requirements relative to full fine-tuning. Past work has shown that LoRA matches full fine-tuning performance for tasks like sequence classification (Hu et al., 2021), instruction tuning (Dettmers et al., 2023; Ghosh et al., 2024), and chat (Dettmers et al., 2023). Other work has shown a gap in the performance of full fine-tuning and LoRA on harder tasks like code generation (Biderman et al., 2024; Zhuo et al., 2024). While we focus on models trained to similar accuracy, our observations of structural differences apply even to cases where LoRA does not fit to the adaptation task as well as full fine-tuning.

**LoRA, formally.** Given a pre-trained weight matrix $W_0 \in \mathbb{R}^{m \times n}$, full fine-tuning treats the learned matrix update as $\Delta W \in \mathbb{R}^{m \times n}$. Instead, LoRA decomposes $\Delta W$ into a product of two matrices such that $\Delta W = BA$, where $B \in \mathbb{R}^{m \times r}$, $A \in \mathbb{R}^{r \times n}$, and where the rank $r$ is generally $r \ll min(m, n)$. During prediction,

$$Y = W_{tuned}X = (W_0 + \frac{\alpha}{r}BA)X .$$

$B$ is initialized to zero, and $A$ sampled from an isotropic Gaussian. All parameters in $B$ and $A$ are trained. From this we can see that while full fine-tuning mas $mn$ trainable parameters per weight matrix, LoRA only has $mr + rn$ trainable parameters. See Appendix D for derivation of gradients.

**LoRA Variants.** Many variations of LoRA exist. Methods improve LoRA's performance or memory-efficiency by initializing with the principal components of the underlying weight matrix (Meng et al., 2024), training with quantization (Dettmers et al., 2023), adaptively allocating different ranks (Zhang et al., 2023), or sequentially training multiple LoRAs (Xia et al., 2024). Other methods propose similar but alternative architectures (Liu et al., 2024; Kopiczko et al., 2024; Koohpayegani et al., 2024). Here, we focus on the original LoRA setup, as described in Hu et al. (2021). We leave a rigorous analysis of these variations and their impacts on our findings to future work.

Empirically, setting $\alpha = 2r$ has been shown to improve results for higher ranks (Biderman et al., 2024) and is theoretically well motivated. (Kalajdzievski, 2023). We adopt this parameterization for most experiments in our paper.

**Analysis of Solutions.** Introduced by Li et al. (2018), the intrinsic dimension measure was used by Aghajanyan et al. (2021) to argue that the fine-tuning update for a pre-trained LLM has low intrinsic rank, explaining why only a small number of trainable parameters are necessary to reach 90% of full

fine-tuning performance. This finding motivated Hu et al. (2021) to hypothesize that LoRA works because solutions of low intrinsic rank exist. But to our knowledge, no past work has compared the rank (or other properties of weight matrices) between LoRA and full-fine tuning on tasks where they are matched in performance. While Liu et al. (2024) showed that LoRA has difficulty changing directional and magnitude components of a neuron independently, while full fine-tuning does not, it is unclear if this difference is due to an inability of LoRA to fit as well as full fine-tuning to the adaptation task.

Recent work comparing LoRA to full fine-tuning has found that LoRA forgets less on previously learned information (Biderman et al., 2024) and more closely resembles the pre-trained model (Ghosh et al., 2024). Surprisingly, some experiments in the current study show opposite trends. However, there are significant differences in the datasets used for evaluation—(Biderman et al., 2024) investigated instruction tuning for language generation, while we mainly study sequence labeling tasks. Importantly, Biderman et al. (2024) study conditions when LoRA fine-tuned models fail to fit the adaptation task as well as full-finetuned models, and as a result also forget less of the pre-training distribution. However, we study models where the LoRA achieves the same performance as full fine-tuning, comparing generalization behavior at a fixed target task accuracy.

**Singular Value Decomposition.** The SVD decomposes a matrix $M \in \mathbb{R}^{m \times n}$ such that $M = U\Sigma V^T$, where $U \in \mathbb{R}^{m \times m}$ and $V \in \mathbb{R}^{n \times n}$ have orthonormal columns representing the singular vectors of $M$ and $\Sigma \in \mathbb{R}^{m \times n}$ is a diagonal matrix containing the singular values of $M$. $U$ and $V^T$ represent rotations that matrix $M$ performs, while $\Sigma$ represents scaling along those axes. Importantly, singular vectors ranked in order by their associated singular value capture the order of most important axes of transformation that the matrix performs.

## 3 MODEL DIFFERENCES BETWEEN LoRA AND FULL FINE-TUNING

Inspired by Sharma et al. (2024)'s findings that the Singular Value Decomposition (SVD, Klema & Laub, 1980) can be used to selectively prune singular vectors to improve model performance, this paper adopts the SVD of neural network parameters as a lens for understanding the changes that fine-tuning makes to pre-trained weights. Understanding how these dimensions change can give us insight into how a particular fine-tuning method changes the pre-trained model. In particular, we measure how well singular vectors in weight matrices fine-tuned with LoRA or full fine-tuning map to singular vectors in the pre-trained weights using their cosine similarity. These relationships are shown in Fig. 1 and Fig. 3, with color representing cosine similarity between pre-trained and fine-tuned singular vectors.

Visually, we observe in Fig. 2(b) that LoRA and full fine-tuning's singular vectors have very different similarities to the pre-trained singular vectors: singular vectors of models fine-tuned with LoRA appear to have, on average, much lower cosine similarity to pre-trained singular vectors in comparison to full fine-tuning. Interestingly, in LoRA fine-tuned models, we also observe the presence of high ranking singular vectors with very low cosine similarity to any pre-trained singular vector.[1] In Fig. 2(c), we show the difference between these vectors with low cosine similarity to the pre-trained singular vectors and normal singular vectors from the fine-tuned weights. This "new" dimension can be seen in Fig. 2(b) as the lone red dot in the bottom left corner. We name these "new" dimensions *intruder dimensions*, which we define formally as follows:

**Definition 1** *A singular vector $y_j$ from the fine-tuned weight matrix $W_{tuned}$ is an **intruder dimension** if and only if $\max_i(cos(y_j, x_i)) < \epsilon$, where $\epsilon$ is a similarity threshold and $x_i$ is a singular vector in $W_0$.*

Examples of intruder dimensions may be seen in Fig. 3. Here, we plot the similarity matrix between the top 10 singular vectors (ranked by singular value) in the pre-trained and fine-tuned matrices. While full fine-tuning appears to have a clear one-to-one mapping, LoRA appears to have its mapping shifted by "blank" columns: these are intruder dimensions, with low cosine similarity to every pre-trained singular vector.

---

[1]Recall that in high dimensions, a vector can have low cosine similarity to a set of orthogonal vectors that span a space; see Appendix C for discussion.

It is important to note that in the case of full fine-tuning, the singular vectors that map to a pre-trained singular vector with high cosine similarity also have similar singular values. From these initial measurements, it appears that LoRA and full fine-tuning have structural differences in the changes they make to the pre-trained weights: while full fine-tuning appears to make small changes to the existing singular vectors and singular values, LoRA introduces new singular vectors that have a large contribution to the norm of the updated parameter matrix.

**Setup.** We study RoBERTa-base (Liu et al., 2019), a pre-trained encoder-only language model, fine-tuned on six different sequence classification tasks. We train these models to similar performance on their respective downstream tasks to study how, at a similar level of performance, fully fine-tuned and LoRA fine-tuned models differ. See Appendix A for more fine-tuning details. We compute the total number of intruder dimensions across these models.

**LoRA fine-tuned models contain high-ranking intruder dimensions while fully fine-tuned models do not.** To quantify the size of the set of intruder dimensions for a specific weight matrix, we use the algorithm described in Fig. 4. Concretely, we first compute the SVD of both the pre-trained and resulting LoRA and full fine-tuned weights. Following that, for each of the top $k$ highest-ranking singular vectors, we measure its maximum cosine similarity with all of the pre-trained singular vectors. If this maximum cosine similarity is less than some threshold $\epsilon$, we classify this singular vector as an intruder dimension. Note that both $k$, the number of fine-tuned singular vectors to examine, and $\epsilon$, the

> **Algorithm: Finding Intruder Dimensions.**
>
> **Input:** Pre-trained weights $W_0$, fine-tuned weights $W_{tuned}$, cosine similarity threshold $\epsilon$, and number of fine-tuned singular vectors to examine $k$.
>
> $[U_0, \Sigma_0, V_0^T] \leftarrow \text{SVD}(W_0)$
> $[U_{\text{tuned}}, \Sigma_{\text{tuned}}, V_{\text{tuned}}^T] \leftarrow \text{SVD}(W_{\text{tuned}})$
> $\text{num\_intruders} \leftarrow 0$
> **for** $j \leftarrow 1$ to $k$ **do**
>    $n \leftarrow$ # of pre-trained singular vectors
>    **if** $\forall i \in \{1, \ldots, n\}, \cos(U_0[i], U_{tuned}[j]) < \epsilon$ **then**
>       $\text{num\_intruders} \leftarrow \text{num\_intruders} + 1$
>    **end if**
> **end for**
> **return** num\_intruders

Figure 4: Outline of the procedure used to compute the total number of intruder dimensions introduced in a model.

cosine similarity threshold, are hyperparameters; we verify the robustness of our findings for a wide range of $\epsilon$ and $k$ values in Fig. 5 and Fig. 11 respectively. To determine the number of intruder dimensions in a specific model, we run this algorithm for each weight matrix in the model and sum the total.

To characterize the differences in fine-tuning methods, we first evaluate the differences in the total number of intruder dimensions in the top 10 highest-ranking singular vectors ($k = 10$). We repeat this procedure for a the range of $\epsilon$ values, our cosine similarity threshold. The results are presented in Fig. 5a. We find that models trained with LoRA consistently contain intruder dimensions when their rank $r \leq 16$, particularly for low values of $\epsilon$. Interestingly, we observe that fully fine-tuned models almost *never* contain intruder dimensions in their top 10 singular vectors for epsilon values of about 0.6 to 0.9 across different settings. This means that full fine-tuning makes smaller changes to the same set of high contribution pre-trained singular vectors. Importantly, the number of intruder dimensions appears to drop as rank increases past a certain threshold, suggesting that the low-rank nature, as well as the update rule of LoRA, induces them to occur.

**Intruder dimensions exist even in tasks where LoRA fine-tuned models learn less than full fine-tuning.** To test the validity of our findings for larger models and harder tasks, we study LLaMA-7B (Touvron et al., 2023a) and LLaMA2-7B (Touvron et al., 2023b) models fine-tuned on various instruction following datasets. These span math, code, and chat, and are considerably harder than our sequence classification tasks. See Appendix H for more details about these models.

Looking at Figs. 5b, 5c, and 5d, we can clearly see intruder dimensions in the set of high ranking singular vectors for LoRA, even with a rank as high as $r = 256$. Importantly, the $r = 2048$ case of MetaMath does not have intruder dimensions and instead has a very similar curve to full fine-tuning. This supports the earlier finding that, as rank increases past a threshold, intruder dimensions disappear and LoRA begin to resemble full fine-tuning.

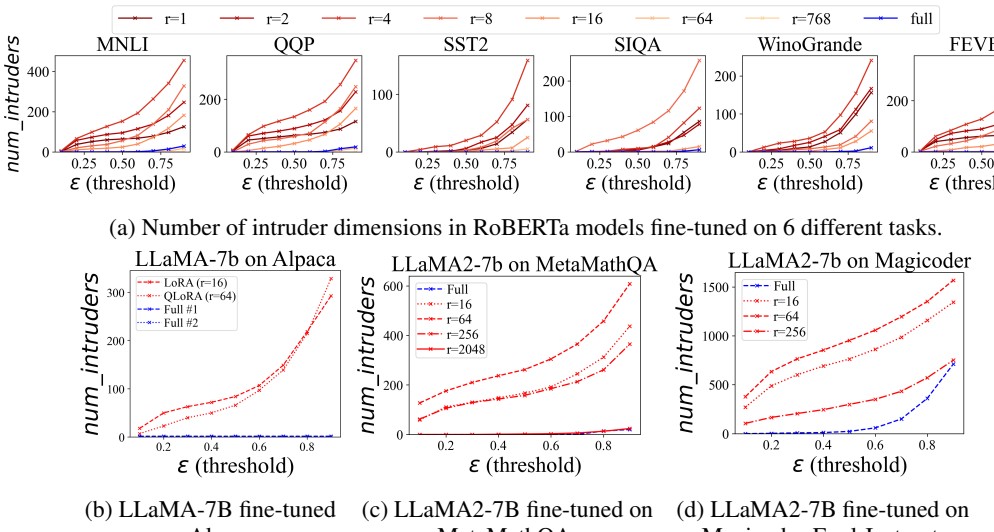

(a) Number of intruder dimensions in RoBERTa models fine-tuned on 6 different tasks.

(b) LLaMA-7B fine-tuned on Alpaca.

(c) LLaMA2-7B fine-tuned on MetaMathQA.

(d) LLaMA2-7B fine-tuned on Magicoder-Evol-Instruct.

Figure 5: **Impact of cosine similarity threshold $\epsilon$ on the number of intruder dimensions.** Here, we set $k = 10$ and measure the impact of $\epsilon$ on the number of intruder dimensions measured. LoRA introduces many intruder dimensions in the top 10 ranking singular vectors, while full fine-tuning does not. Top row is for RoBERTa-base. Numbers are reported for the entire model, so upper bound is $k * l * n$, where $l$ is the number of layers and $n$ is the number of weight matrices per layer. For RoBERTa-base, this upper bound is $10 * 6 * 12 = 720$.

Interestingly, the full fine-tuned Magicoder model also has intruder dimensions for higher values of $\epsilon$. This is likely because, as mentioned by Biderman et al. (2024), there is a larger domain shift between coding tasks and the pre-training data in comparison to other natural language tasks. This difference likely causes full fine-tuning to make more aggressive changes to the model. But even in this case, LoRA models have many more intruder dimensions in their top 10 singular vectors than full fine-tuning (see Fig. 1).

**Full fine-tuning updates have a higher effective rank than LoRA updates, even when LoRA is performed with a full-rank matrix.** Another way we can examine differences between LoRA and full fine-tuning is to calculate the effective rank (Roy & Vetterli, 2007) of the change made to the weights during fine-tuning. As shown in Fig. 6, when we calculate this we observe that the effective rank of full fine-tuning solutions have a significantly higher effective rank than solutions learned by LoRA, even when LoRA has high rank. Even at high adapter ranks and with rank stabilization, we find across layers that the effective rank of LoRA updates is less than half that of full fine-tuning and a quarter of the adapter rank. For example, with the high rank of $r = 768$ for RoBERTa, LoRA updates have an average effective rank of 300. This suggests that LoRA is under utilizing its full capacity $r$, and may help explain observed gaps between LoRA and full fine-tuning on challenging tasks like coding (Biderman et al., 2024; Zhuo et al., 2024).

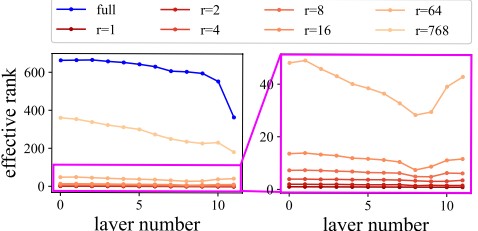

Figure 6: **Very high rank LoRA updates still have lower effective rank than full-finetuning.** This holds across all matrix types and layers. *(Left)* Effective rank LoRA and Full. *(Right)* Zoomed in on only LoRA.

**Intruder dimensions are distributed across both high and low singular values.** We examine the extent to which intruder dimensions exist throughout the entire weight matrix and how they are distributed. To do this, we hold $\epsilon$ fixed and measure the number of intruder dimensions while varying the proportion of the fine-tuned singular vectors that we examine. We report these results in Fig. 11a. Here, we can see that LoRA consistently has more intruder dimensions than full fine-tuning, regardless of what fraction of the singular values we examine. The only caveat to this is

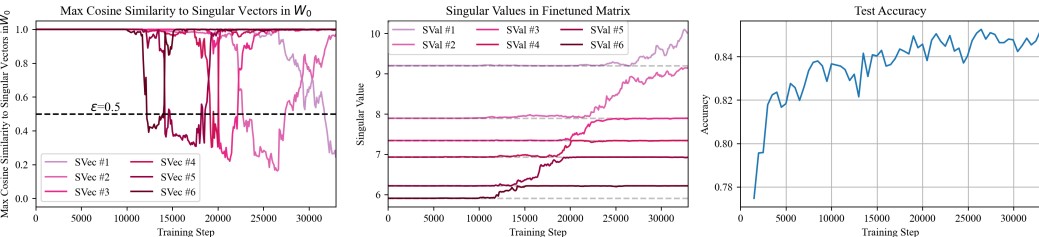

Figure 7: **Evolution of the intruder dimension with training iterations.** *(Left)* Intruder dimensions, and their rank, in a LoRA fine-tuned weight matrix during fine-tuning. *(Middle)* Their associated singular values. This clearly shows that across training steps, the impact of the intruder dimension, as determined by its singular value, increases. *(Right)* Test accuracy of the model across training steps.

that, for some datasets, full fine-tuning passes LoRA with rank 1 when examining the last 20% of the fine-tuned singular vectors. This is likely due to the limited expressivity of rank 1 updates and is interesting because it suggests that in these cases, full fine-tuning may be changing lower-ranking singular vectors more than LoRA.

**Intruder dimensions increase in magnitude and change in direction as fine-tuning progresses.** To further understand how a particular intruder dimension is introduced during fine-tuning with LoRA, we measure the maximum cosine similarity between the top individual fine-tuned singular vectors and all the pre-trained singular vectors across many intermediate steps in the fine-tuning process, as seen in Fig. 7 (*left*). In parallel, we track changes in their associated singular values as seen in Fig. 7 (*right*). As is evident from the graphs, intruder dimensions appear to gradually increase their "rank" (on the left) as their singular value is increased (on the right) while simultaneously changing in direction too as training progresses.

**Scaling $\alpha$ with the rank of the LoRA update reduces the number of intruder dimensions alongside increasing the effective ranks of the matrices.** Following (Biderman et al., 2024), we set $\alpha = 2r$. However, we ran additional experiments with a fixed $\alpha = 8$, as in most early work on LoRA. This has the effect of scaling down the LoRA contribution as rank increases. We report these results in Appendix 18. For both settings of $\alpha$, models obtained equivalent performance on the target task (see Table 1). With fixed $\alpha$, however, all ranks of LoRA—even very large ones—exhibit intruder dimensions. Furthermore, when we measure the effective rank of these models, they have a much smaller effective rank than when $\alpha = 2r$. This suggests that with constant $\alpha$, LoRA *converges to a low rank solution*. This provides additional evidence that $\alpha = 2r$ improves the solution of high ranks of LoRA(Kalajdzievski, 2023; Biderman et al., 2024): it leads to a reduction in intruder dimensions and an increase in the effective rank of solutions when LoRA's rank is higher.

**The total number of intruder dimensions increases proportionally to the size of the fine-tuning dataset.** Using the training described in Appendix A, we fine-tuned models on data subsets of varying sizes. We trained RoBERTa-base on MNLI using LoRA with rank 1 and 8 (cases where we originally saw intruder dimensions). We then again measure number of intruder dimensions along with the impact of $\epsilon$ and $k$, and report our results in Appendix 12. For $r = 8$, as we train on more data, more intruder dimensions are introduced. Interestingly, however, LoRA with rank 1 appears to converge to similar amounts of intruder dimensions, regardless of the dataset size. This may be because of the limited expressivity of models with $r = 1$.

**Conjecture: Intruder dimensions, as high-ranking singular vectors, contribute significantly to the norm and stability of the parameter matrix.** In contrast to pre-trained singular vectors that are learned from large pre-training corpora, LoRA introduces intruder dimensions learned solely from the smaller dataset of the fine-tuning task, which overpower the pre-trained vectors, as seen in the experiments so far. On the other hand, full fine-tuning, while adapting just as effectively to the fine-tuning task, retains the spectral properties of the pre-trained model effectively. From this, we conjecture that the presence of intruder dimensions in LoRA models has a detrimental effect on the model's performance outside the fine-tuning task distribution and this effect is less pronounced in full fine-tuned models. We investigate this conjecture in the next section.

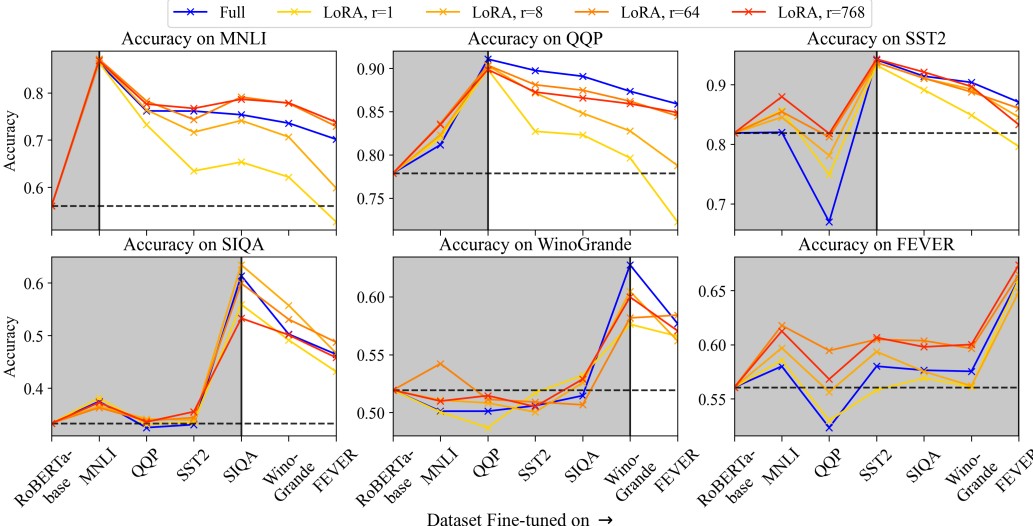

Figure 8: Continual Learning performance of RoBERTa for full fine-tuning and LoRA. We sequentially train on six tasks, in order from left to right. Horizontal dotted line indicates baseline pre-trained performance. Vertical solid line indicates when a specific dataset is fine-tuned on. Gray region represents performance before the model has been trained on that task. We are interested in the differences in accuracies of these methods both right after training (at the vertical black line) and later (in the white region). We see that low ranks of LoRA forget previously learned tasks more.

## 4 BEHAVIORAL DIFFERENCES BETWEEN LORA AND FULL FINE-TUNING

We have identified structural differences in the solutions of LoRA and full fine-tuning. Here, we investigate whether LoRA and full fine-tuning produce measurable differences in fine-tuned model behavior. While we have already seen that they perform nearly identically on their in-distribution test set, we evaluate whether these behavioral similarities hold under other distributions.

**At lower ranks, LoRA adapts less robustly during continual learning by forgetting more of the previous tasks.** We train RoBERTa sequentially on multiple tasks and measure how much performance changes as new tasks are learned. We use the same training recipe and datasets as before, but now instead fine-tuning in a continual learning environment with the following dataset order: MNLI, QQP, SST-2, SIQA, Winogrande, FEVER. After training on a certain dataset in the sequence, we merge the LoRA weights into the model and reinitialize them before training on the next task so that they are unimpacted by the previous tasks. After training on a specific task, we test on all tasks by, for each task, separately retraining its classification head before testing on its test set. This enables us to examine how well the model performs on these tasks while not actually changing the model itself.

Results are shown in Fig. 8. While LoRA matches the performance of full fine-tuning initially, smaller ranks of LoRA consistently show greater degradation of performance during continual learning. In particular, we note that for the first three datasets trained on, performance of LoRA when $r = 1$ *drops below the pre-trained baseline.* As we increase the rank of LoRA, we can see that this forgetting behavior decreases and more closely resembles full fine-tuning and even forgets less on MNLI after the completion of continual learning. Biderman et al. (2024) describe a family of tasks and training procedures under which LoRA forgets less than full fine-tuning, these results show that the complete picture is nuanced: while in some cases LoRA appears to forget less, for some tasks—and some ranks—LoRA may in fact forget more.

**For LoRA models fine-tuned to equivalent test accuracy, we see a U-shaped curve that identifies the optimal rank for fitting to the downstream task while forgetting the pre-training distribution the least.** We measure the shift in performance that our fine-tuned models, trained to equivalent test accuracy, have on their pre-training data distribution. While we cannot directly measure a true perplexity on encoder-only style models because they are not auto-regressive language

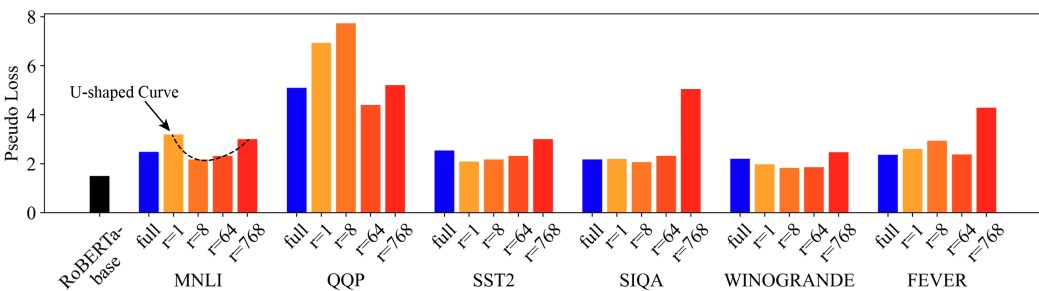

Figure 9: RoBERTa's performance on its pre-training data distribution after fine-tuning on a particular task. We measure pseudo loss as described by Salazar et al. (2020). All the models for a specific task were trained to equivalent performance. We see a U-shaped curve that identifies the best rank for learning the downstream task while forgetting the pre-training distribution the least.

models, we can still measure their pseudolikelihood on pre-training data as described in Salazar et al. (2020). We measure "pseudo-loss" for all our fine-tuned RoBERTa models across the four datasets that RoBERTa used during pre-training (openwebtext (Gokaslan & Cohen, 2019), cc_news (Hamborg et al., 2017), stories (Trinh & Le, 2019), and bookcorpus (Zhu et al., 2015)), and weigh them proportionally to their contribution as described by Liu et al. (2019). We report our measured pseudo-loss scores in Fig. 9. In it, we can see a U-shaped trend between full fine-tuning and LoRA with $r = 768$. Since all models achieve equivalent test accuracy, this U-shaped trend across a specific dataset identifies the optimal ranks for fitting to a down stream task distribution, and seems to point to $r = 64$ as the choice that minimizes forgetting of the pre-training distribution. We can see that both a rank very low rank ($r = 1$) and a very high rank ($r = 768$) lead to greater forgetting on the pre-training distribution relative to full fine-tuning, while for $r = 64$ we see less. That is: models fine-tuned with LoRA when $r = 1$ suffer from intruder dimensions and appear to have more forgetting than $r = 64$ which had no intruder dimensions. However, models fine-tuned with LoRA when $r = 768$ also exhibit worse forgetting, suggesting that due to their overparameterization they are overfitting to the adaptation task. With $r = 8$ and $r = 64$, which are more frequently used, forget less than full fine-tuning, while ranks on either extreme forget more than full fine-tuning.

**Setting $\alpha$ properly significantly impacts model performance.** We continue our case study of setting $\alpha = 8$ instead of $\alpha = 2r$ as described in earlier sections. We repeat continual learning and pre-training forgetting experiments with fixed (rather than rank-scaled) $\alpha$, and report them in Appendix 16 & 17. LoRA models, regardless of rank, forget much more of both the pre-training distribution (MNLI, QQP, FEVER) and previously learned tasks during continual learning, highlighted by the fact that *all LoRA ranks drop below baseline performance for the first two datasets*. These results resemble earlier findings for $r = 1$, and further suggests that when $\alpha = 8$ instead of $\alpha = 2r$, solutions converge to a solution with more intruder dimensions structurally and one that is behaviorally similar to the low-rank LoRA setting.

## 5 WHY DO INTRUDER DIMENSIONS EXIST?

**Adding an random vector to a pre-trained matrix introduces an intruder dimension:** To help provide intuition about how new singular vectors in the SVD can be added by LoRA, we examine mathematical conditions that lead to their creation. Certainly, when comparing $SVD(W + \lambda vv^T)$ and $SVD(W)$, where $W$ are the pre-trained weights in $\mathbb{R}^{n \times n}$, $v$ is a randomly sampled vector in $\mathbb{R}^n$, and $\lambda$ is a scalar value greater than the largest singular value of $W$, we expect this update to create an intruder dimension (as $v$ is nearly orthogonal to the existing singular vectors w.h.p.).

**Differences in the update rule:** As described in Appendix D, LoRA and full fine-tuning have characteristically different update rules, even for the same training examples. We highlight that LoRA uses a larger learning rate and has gradients projected into a low-rank space (Hao et al., 2024), leading to conditions similar to the toy example above.

**Product parameterization of LoRA:** Multiplying matrices together amplifies their spectral differences (their singular values) and in most cases leads to a lower effective rank. To test the impact of

Figure 10: **Impact of only tuning B on the number of intruder dimensions.** We randomly initialize A such that it has singular values of 1, freeze it, and only train B. When we do this, we see a sharp reduction in high ranking intruder dimensions in comparison to those in normal LoRA (reported in Fig. 5a). Graphs for a specific dataset have the same range as Fig. 5a for easy comparison.

the product $BA$ on the introduction of intruder dimensions, we randomly initialize $A$ such that all its singular values are 1 and freeze it. We only tune $B$ and keep the rest of our fine-tuning recipe the same. Comparing this with vanilla LoRA is fair because Zhu et al. (2024) found that tuning $B$ is more impactful and important for generalization in comparison to $A$ and Hao et al. (2024) showed that only tuning $B$ effectively approximates LoRA. As we can see in Fig. 10, we see a sharp drop in the number of high ranking intruder dimensions when only tuning $B$ in comparison to the vanilla LoRA case where we train $A$ and $B$ separately, as reported in Fig. 5. This suggests that the matrix product of LoRA is an important component in the introduction of intruder dimensions because of how it amplifies the spectral differences of $B$ and $A$.

## 6 CONCLUSION

The paper describes the finding that LoRA and full fine-tuning, with equal performance on the fine-tuning task, can have solutions with very different generalization behaviors outside the fine-tuning task distribution. We found that LoRA and full fine-tuning yield models with significant differences spectral properties of their weight matrices: LoRA models often containing "intruder dimensions", high-ranking singular vectors approximately orthogonal to the singular vectors of pre-trained weight matrices. The existence of intruder dimensions correlates with the fine-tuned model forgetting more of the pre-training distribution as well as forgetting more when trained on tasks sequentially in a continual learning setup.

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

## A ROBERTA FINE-TUNING DETAILS

We generally follow the procedure used by Hu et al. (2021). For all models, we use a linear learning rate schedule with 0.06 linear warmup ratio and train for a maximum of 5 epochs with batch size 16. We use the Adam optimizer with no weight decay and a maximum sequence length of 512. We fine-tune all linear layers besides the embedding matrix as well as all bias and LayerNorm layers to ensure fair comparison between methods. For full fine-tuning, we use a learning rate of 1e-5. For LoRA, we set $\alpha = 2r$, and train for all ranks in $\{1, 2, 4, 8, 16, 64, 768\}$. We hold the "total learning rate of LoRA", which is $\alpha * \eta$, fixed as we sweep rank such that this product always equals 2.4e-3. We train these models to equivalent accuracy on their downstream task. We fine-tune on six sequence classification tasks: sentiment analysis (Socher et al., 2013), entailment (Williams

et al., 2018), duplicate identification (Wang et al., 2019), fact verification (Thorne et al., 2018), and common sense reasoning (Sap et al., 2019; Sakaguchi et al., 2021).

## B    MODEL ACCURACIES

We provide the accuracies that our RoBERTa models achieve in Table 1 and Table 2.

| Model | Type | MNLI | SST-2 | QQP | WinoGrande | SIQA | FEVER |
|-------|------|------|-------|-----|------------|------|-------|
| | Full | 0.8617 | 0.9461 | 0.9146 | 0.6251 | 0.6551 | 0.6687 |
| | r=1 | 0.8647 | 0.9358 | 0.9045 | 0.6251 | 0.672 | 0.6712 |
| | r=2 | 0.8604 | 0.9415 | 0.9058 | 0.6172 | 0.6581 | 0.6673 |
| $\text{RoB}_{base}$ | r=4 | 0.8607 | 0.9369 | 0.9079 | 0.6472 | 0.6505 | 0.6694 |
| | r=8 | 0.8648 | 0.9438 | 0.9108 | 0.6417 | 0.6586 | 0.6582 |
| | r=16 | 0.8604 | 0.9427 | 0.9095 | 0.6235 | 0.6853 | 0.663 |
| | r=64 | 0.8671 | 0.9484 | 0.9117 | 0.6614 | 0.6638 | 0.6601 |
| | r=768 | 0.8694 | 0.9369 | 0.9118 | 0.6361 | 0.6607 | 0.6641 |

Table 1: Model accuracies on their given downstream task after fine-tuning for $\alpha = 8$.

| Model | Type | MNLI | SST-2 | QQP | WinoGrande | SIQA | FEVER |
|-------|------|------|-------|-----|------------|------|-------|
| | Full | 0.8617 | 0.9461 | 0.9146 | 0.6251 | 0.6551 | 0.6687 |
| | r=1 | 0.8615 | 0.9427 | 0.9033 | 0.6212 | 0.6305 | 0.6794 |
| | r=2 | 0.8639 | 0.9392 | 0.9053 | 0.6369 | 0.6530 | 0.6663 |
| $\text{RoB}_{base}$ | r=4 | 0.8615 | 0.9438 | 0.9083 | 0.6440 | 0.6633 | 0.6667 |
| | r=8 | 0.8707 | 0.9415 | 0.9079 | 0.6322 | 0.6571 | 0.6739 |
| | r=16 | 0.8666 | 0.9495 | 0.9088 | 0.6338 | 0.6679 | 0.6730 |
| | r=64 | 0.8710 | 0.9473 | 0.9073 | 0.6283 | 0.6274 | 0.6780 |
| | r=768 | 0.8690 | 0.9381 | 0.9024 | 0.6133 | 0.6274 | 0.6729 |

Table 2: Model accuracies on their given downstream task after fine-tuning for $\alpha = 2r$.

## C   Cosine Similarity with Orthogonal Vectors that Span a Space

Here we demonstrate why it is possible for a vector to have low cosine similarity with every orthogonal vector that collectively span a space if the dimensionality of the vectors is high.

**Minimizing the Maximum Cosine Similarity.** Lets take $Z = \min\limits_{v \in \mathbb{R}^n} \max\limits_{i} cos(v, x_i)$, where $v$ is an arbitrary vector and each vector $x_i$, which we collectively call $X$, make up an orthonormal basis that span the space. $Z$ can be small in a high dimensional space.

**2-D case.** Assume $X = I$ without loss of generality. It is trivial to see that $Z = \frac{1}{\sqrt{2}}$, and is when $v = \begin{bmatrix} \frac{1}{\sqrt{2}} & \frac{1}{\sqrt{2}} \end{bmatrix}$.

**3-D case.** Assume $X = I$ without loss of generality. $Z = \frac{1}{\sqrt{3}}$ when $v = \begin{bmatrix} \frac{1}{\sqrt{3}} & \frac{1}{\sqrt{3}} & \frac{1}{\sqrt{3}} \end{bmatrix}$.

**N-D case.** In the N-D case, we can see, via induction, that $Z = \frac{1}{\sqrt{n}}$.

As we can see here, if $n$ is large, the value of $Z$ will be low, even though we are doing the cosine similarity of a vector with respect to a set of orthonormal vectors that span a space.

# D DERIVATION OF GRADIENTS

Our calculations follow a similar line to that of Hao et al. (2024).

**Derivation for Full Fine-tuning.** Full fine-tuning is structured such that

$$Y = W_{tuned}X = (W_0 + \Delta W)X,$$

where $X \in \mathbb{R}^{n \times b}$ are the inputs, $Y \in \mathbb{R}^{m \times b}$ are the outputs, $W_0 \in \mathbb{R}^{m \times n}$ are the pre-trained weights, and $\Delta W \in \mathbb{R}^{m \times n}$ is the fine-tuning update. Accordingly, $\frac{\partial L}{\partial \Delta W} = \frac{\partial L}{\partial Y}X^T$, and the update is

$$\Delta W_n = \Delta W_{n-1} - \eta \frac{\partial L}{\partial Y}_n X_n^T,$$

where $\eta$ is the learning rate.

**Derivation for LoRA.** LoRA is structured such that

$$Y = W_{tuned}X = (W_0 + \frac{\alpha}{r}BA)X,$$

where $X \in \mathbb{R}^{n \times b}$ are the inputs, $Y \in \mathbb{R}^{m \times b}$ are the outputs, $W_0 \in \mathbb{R}^{m \times n}$ are the pre-trained weights, $B \in \mathbb{R}^{m \times r}$ is initialized to zero, $A \in \mathbb{R}^{r \times n}$ is randomly initialized, and $\alpha$ is a hyperparameter. Accordingly, $\frac{\partial L}{\partial B} = \frac{\alpha}{r}\frac{\partial L}{\partial Y}X^T A^T$ and $\frac{\partial L}{\partial A} = \frac{\alpha}{r}B^T\frac{\partial L}{\partial Y}X^T$. Therefore, their respective updates are

$$B_n = B_{n-1} - \eta \frac{\alpha}{r}\frac{\partial L}{\partial Y}X^T A^T$$

and

$$A_n = A_{n-1} - \eta \frac{\alpha}{r}B^T\frac{\partial L}{\partial Y}X^T,$$

where $\eta$ is the learning rate.

**Differences in First Step.** During the very first step of training, given identical examples both full fine-tuning and LoRA have the same $X$ and $Y$ for each layer since $B$ is initialized to zero. After the first step, full fine-tuning has a update matrix equal to

$$\Delta W_{full} = -\eta \frac{\partial L}{\partial Y}X^T.$$

In contrast, LoRA has an update matrix equal to

$$\Delta W_{lora} = (\frac{\alpha}{r})(B_0 - \eta \frac{\alpha}{r}\frac{\partial L}{\partial Y}X^T A_0^T)(A_0 - \eta \frac{\alpha}{r}B_0^T\frac{\partial L}{\partial Y}X^T).$$

Since $B_0 = 0$,

$$\Delta W_{lora} = (\frac{\alpha}{r})(-\eta \frac{\alpha}{r}\frac{\partial L}{\partial Y}X^T A_0^T)(A_0).$$

From this, we can see that the gradient steps are clearly different, even with the same training examples.

# E  IMPACT OF MATRIX PERCENTAGE ON NUMBER OF INTRUDER DIMENSIONS

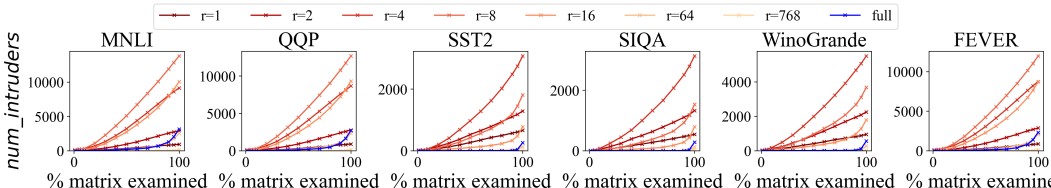

(a) Impact of the number of singular vectors in the fine-tuned matrix we examine, $k$, on the number of intruder dimensions for RoBERTa models fine-tuned on 6 different tasks. Here, we set $\epsilon = 0.5$.

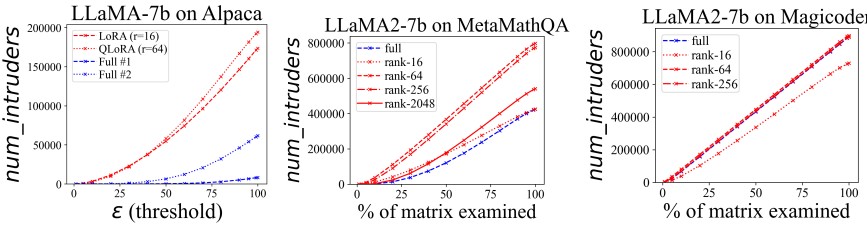

(b) LLaMA-7B fine-tuned on Alpaca.

(c) LLaMA2-7B fine-tuned on MetaMathQA.

(d) LLaMA2-7B fine-tuned on Magicoder-Evol-Instruct.

Figure 11: **Impact of $k$, the number of fine-tuned singular vectors we examine, on the number of intruder dimensions.** We see that models fine-tuned with LoRA tend to have more intruder dimensions than full fine-tuning, regardless of the value of $k$ used.

# F  PLOTS OF IMPACT OF DATASET SIZE

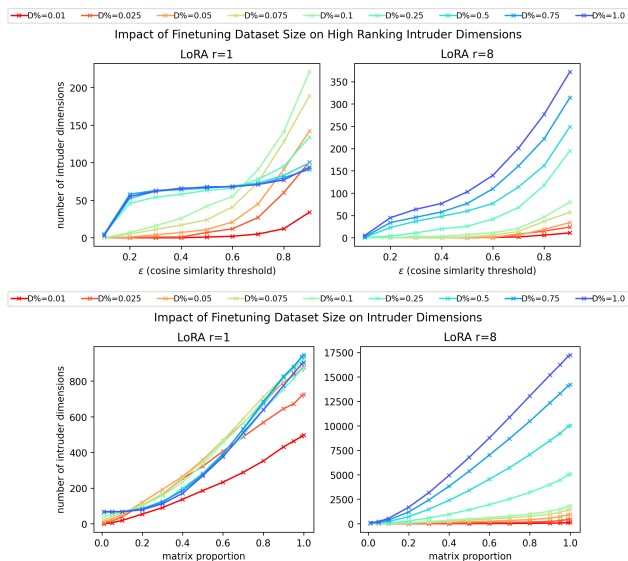

Figure 12: *(Top)* Impact of cosine similarity threshold, $\epsilon$, on the number of intruder dimensions for LoRA models trained on different proportions of the MNLI dataset. *(Bottom)* Impact of the number of fine-tuned singular vectors we examine, $k$, on the number of intruder dimensions for LoRA models trained on different proportions of the MNLI dataset. We see that training on a larger proportion of the dataset increases the number of intruder dimensions in the model.

## G    EFFECTIVE RANK WHEN ALPHA=2R

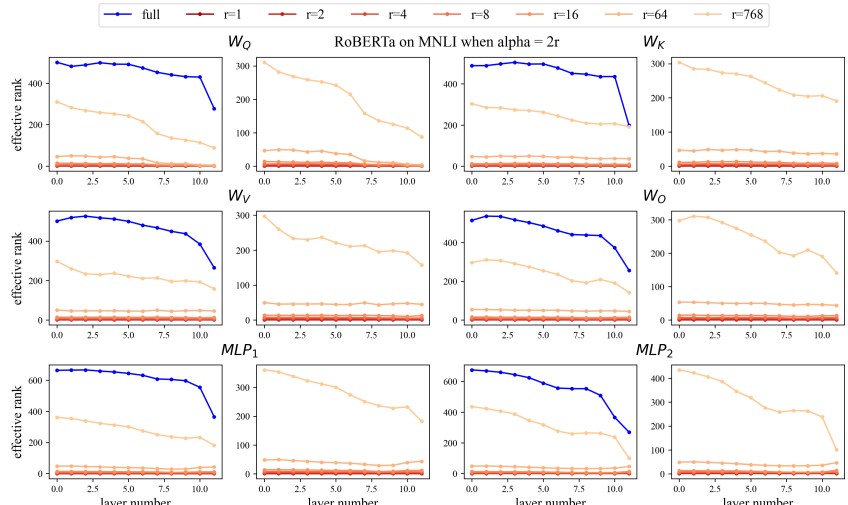

Figure 13: Effective Rank of the update to RoBERTa on MNLI when $\alpha = 2r$. We measure for all weight types. For a specific weight type, the graph on the left shows the effective rank of all models, and the right shows the effective rank of the LoRA models only.

## H    LLAMA/LLAMA-2 INSTRUCTION TUNED MODELS

Our LLaMA-7B checkpoints were fine-tuned on the Alpaca (Taori et al., 2023) and consist of two fully fine-tuned models, one LoRA model with rank 16, and one QLoRA (Dettmers et al., 2023) model with rank 64. Our LLaMA2-7B checkpoints were fine-tuned on either Magicoder-Evol-Instruct-110K (Wei et al., 2024) or MetaMathQA (Yu et al., 2024) and consist of one fully fine-tuned model and 3-4 LoRA'ed models of different ranks for each dataset and generously provided by Biderman et al. (2024).

| Hugging Face Path | Base Model | IT Dataset |
|---|---|---|
| timdettmers/qlora-alpaca-7b | LLaMA-7b | Alpaca |
| tloen/alpaca-lora-7b | LLaMA-7b | Alpaca |
| PKU-Alignment/alpaca-7b-reproduced | LLaMA-7b | Alpaca |
| chavinlo/alpaca-native | LLaMA-7b | Alpaca |
| LoRA-TMLR-2024/magicoder-lora-rank-16-alpha-32 | LLaMA2-7b | Magicoder |
| LoRA-TMLR-2024/magicoder-lora-rank-64-alpha-128 | LLaMA2-7b | Magicoder |
| LoRA-TMLR-2024/magicoder-lora-rank-256-alpha-512 | LLaMA2-7b | Magicoder |
| LoRA-TMLR-2024/magicoder-lora-rank-2048-alpha-4096 | LLaMA2-7b | Magicoder |
| LoRA-TMLR-2024/magicoder-full-finetuning-lr-5e-05 | LLaMA2-7b | Magicoder |
| LoRA-TMLR-2024/magicoder-lora-rank-16-alpha-32 | LLaMA2-7b | MetaMath |
| LoRA-TMLR-2024/magicoder-lora-rank-64-alpha-128 | LLaMA2-7b | MetaMath |
| LoRA-TMLR-2024/magicoder-lora-rank-256-alpha-512 | LLaMA2-7b | MetaMath |
| LoRA-TMLR-2024/magicoder-full-finetuning-lr-1e-05 | LLaMA2-7b | MetaMath |

Table 3: Hugging Face model paths for LLaMA-7b/LLaMA2-7b IT models.

# I CASE STUDY: SETTING ALPHA=8 INSTEAD OF ALPHA=2R

Our main experiments were conducted with $\alpha = 2r$. However, Hu et al. (2021) instead set $\alpha = 8$ for RoBERTa-base. While not the recommended practice now, we explore what impact this selection has on our findings. We report our key plots in Fig. 14, 15, 16, 17, & 18.

In Fig. 14 & 15 we see that LoRA'd models with high rank have significantly more intruder dimensions in comparison to when $\alpha = 2r$. Interestingly, whereas when $\alpha = 2r$ LoRA models with ranks like 64 had no or very few intruder dimensions (see Fig. 5), they now have numerous intruder dimensions.

These differences are corroborated by Fig. 18, where we see that the learned solutions of LoRA have significantly lower effective rank in comparison to when $\alpha = 2r$. For example, we see in Fig. 18 that when LoRA has a rank of 768, the effective rank is never above 100. In contrast, we see in Fig. 13 that with the same rank of 768, LoRA always has an effective rank above 768. This suggests that when $\alpha = 8$, LoRA is converging to lower rank solutions than when $\alpha = 2r$. This supports the finding that setting $\alpha = 2r$ improves LoRA's performance when a high rank is used (Biderman et al., 2024; Kalajdzievski, 2023).

Behaviorally, we see in Fig. 17 that LoRA models with high rank have much more forgetting on previously learned tasks in comparison to full fine-tuning and LoRA when $\alpha = 2r$ is used ($\alpha = 2r$ results are in Fig. 8). Likewise, in Fig. 18 we see that when LoRA has high rank, it has much more forgetting on the pre-trained distribution in comparison to LoRA when $\alpha = 2r$.

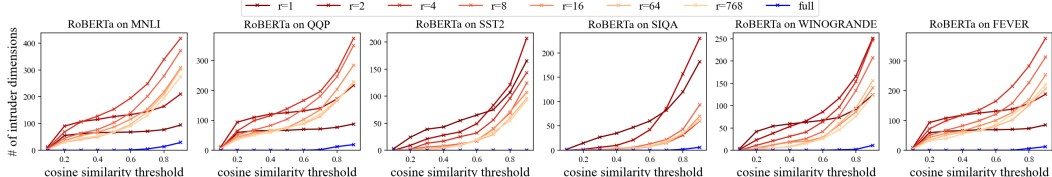

Figure 14: Number of intruder dimensions in RoBERTa models fine-tuned on 6 different tasks. Here, we set $k = 10$. We use the same conditions as in Fig. 5a but instead now set $\alpha = 8$ instead of $\alpha = 2r$.

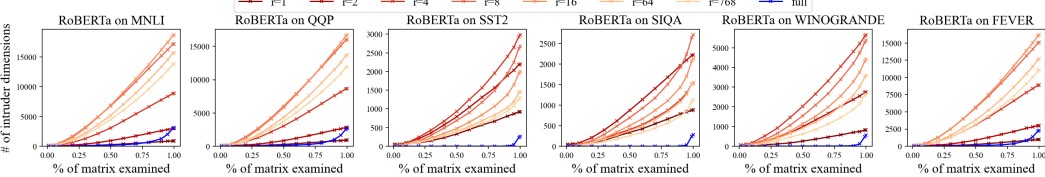

Figure 15: Impact of the number of singular vectors in the fine-tuned matrix we examine, $k$, on the number of intruder dimensions for RoBERTa models fine-tuned on 6 different tasks. Here, we set $\epsilon = 0.5$. We use the same conditions as in Fig. 11a but instead now set $\alpha = 8$ instead of $\alpha = 2r$.

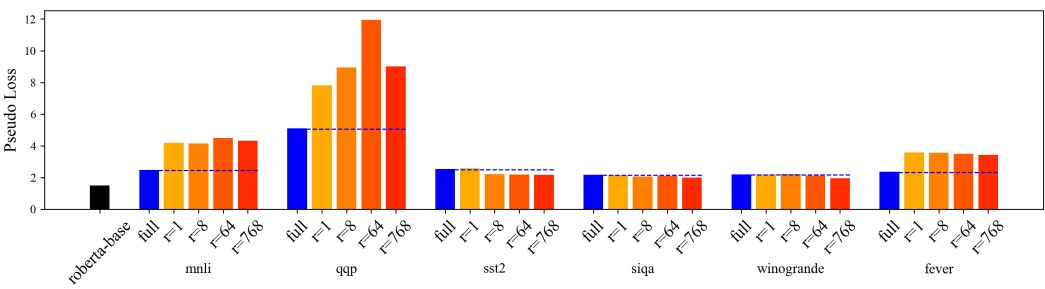

Figure 16: For $\alpha = 8$. RoBERTa's performance on its pre-training data distribution after fine-tuning on a particular task. We measure pseudo loss as described by Salazar et al. (2020). We compare these results to when $\alpha = 2r$ (Fig. 9).

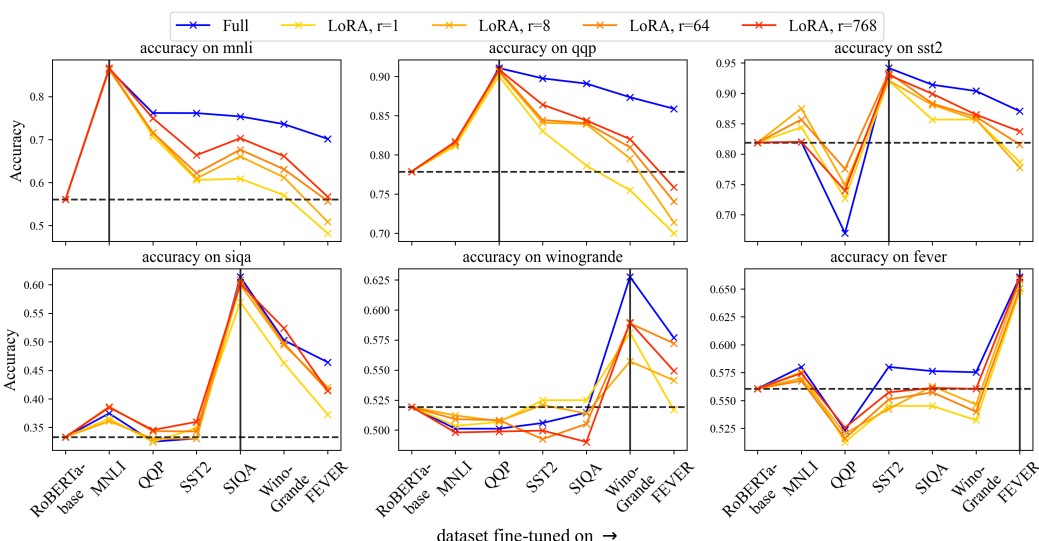

Figure 17: For $\alpha = 8$. RoBERTa's performance on six datasets during continual learning. We sequentially train on six tasks, in order from left to right. Horizontal dotted line indicates baseline pre-trained performance. Vertical solid line indicates when a specific dataset is fine-tuned on. We compare these results to when $\alpha = 2r$ (Fig. 8).

## J  LoRA VARIANTS

## K  IMPACT OF HYPERPARAMETERS

## L  IMPACT OF RANDOM SEEDS

## M  INTRUDER DIMENSIONS CAUSE WORSE OOD PERFORMANCE

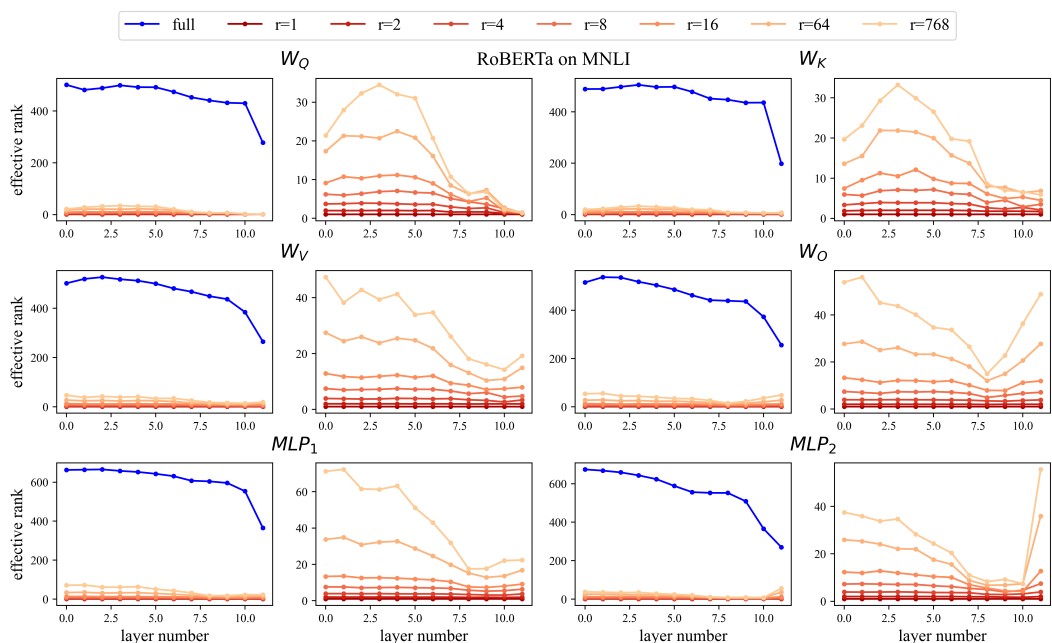

Figure 18: Effective rank of the update to RoBERTa on MNLI when $\alpha = 8$. We compare these results to when $\alpha = 2r$ (Fig. 13). We show for all weight types. For a specific weight type, the graph on the left shows the effective rank of all models, and the right shows the effective rank of the LoRA models only.

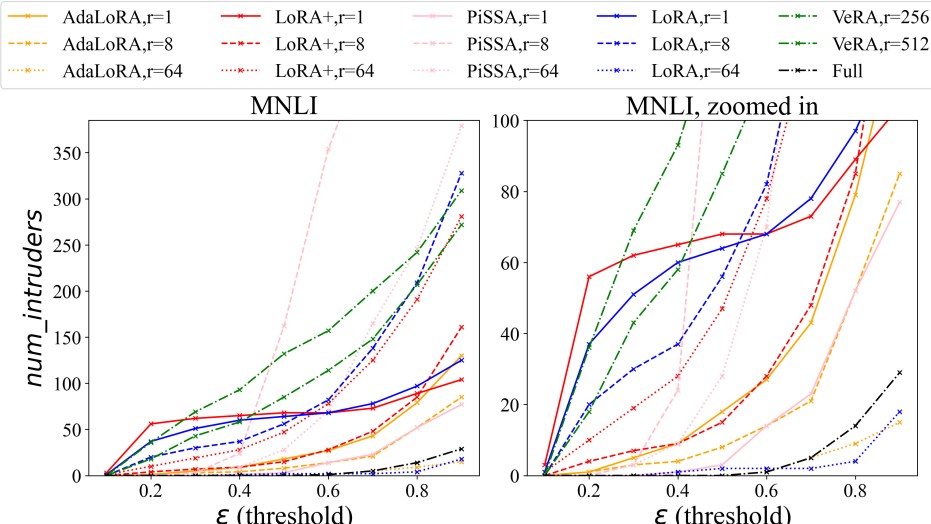

Figure 19: **Intruder Dimension Measurements of LoRA Variants.** We use the same methodology as in Fig. 5, but in addition study AdaLoRA, LoRA+, PiSSA, and VeRA. We find several things from this analysis of the intruder dimensions of various methods. We still find that using a higher rank is effective for reducing the number of intruder dimensions after fine-tuning. Importantly, using a low rank still appears to be a very strong indicator of the presence of intruder dimensions. However, certain methods appear to have fewer in comparison to others: AdaLoRA, which reparametrizes the LoRA update as an SVD-like module, appears to have fewer intruder dimensions, suggesting that this methodology of separating the rotational and scaling components may be beneficial for reducing intruder dimensions.

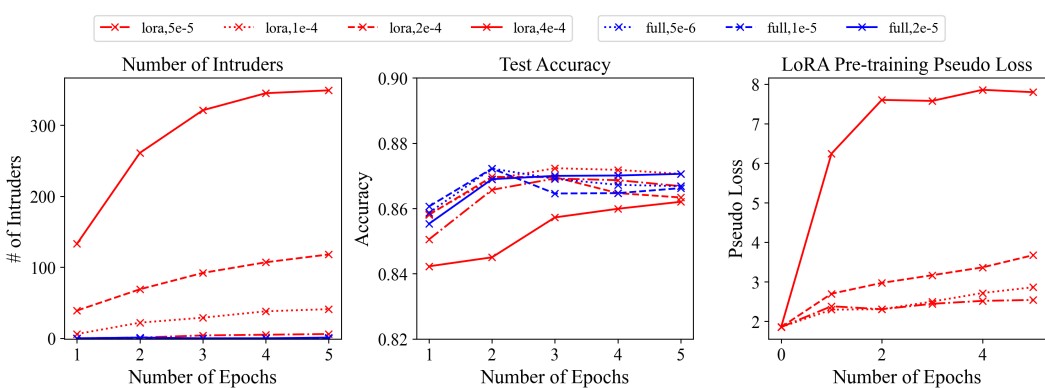

Figure 20: **Impact of Learning Rate and Number of Epochs on Intruder Dimensions and Performance.** We use the same setup as as Fig. 5a with $k = 10$ and $\epsilon = 0.5$ and measure the number of intruder dimensions in the entire model, the model's test accuracy, and the model's pre-training pseudo loss across training epochs for different learning rates. Here, we see that learning rate plays an important role in the introduction of intruder dimensions, with larger learning rates introducing many more intruder dimensions. We also see a clear correlation($\rho = 0.944$, p-value $\leq 0.001$) between number of intruder dimensions and pre-training pseudo loss: more intruder dimensions imply worse OOD performance.

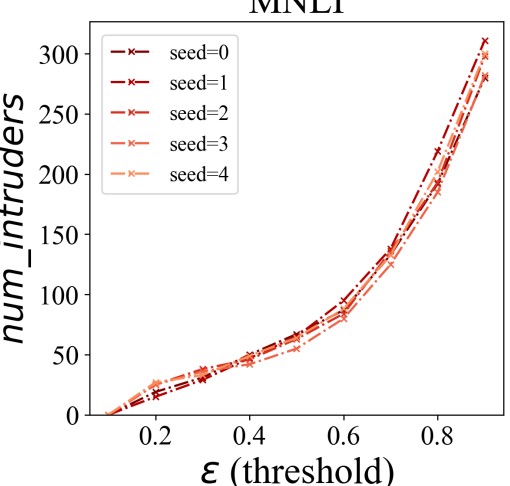

Figure 21: **Impact of Random Seeds on Intruder Dimensions.** We fine-tune RoBERTa-base across 5 random seeds and use our same methodology as in Fig. 5a. We find that the initialization has a negligible role on the number of intruder dimensions.

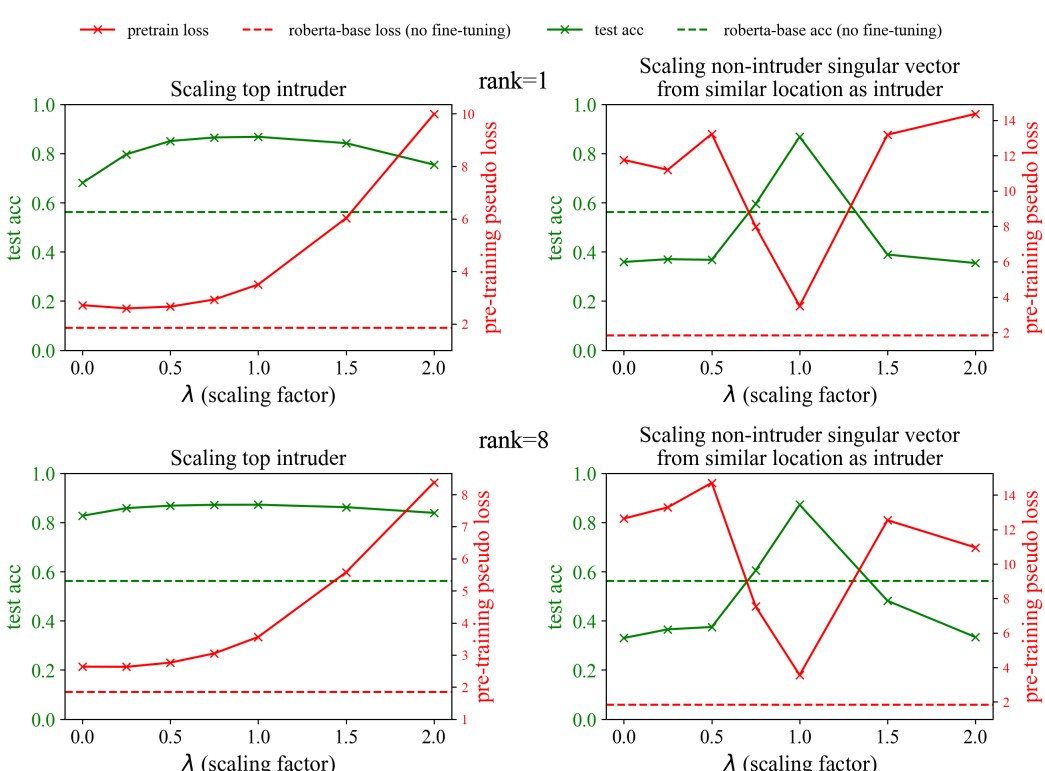

Figure 22: **Intruder Dimensions are responsible for worse OOD performance.** To test the impact of intruder dimensions, we search for the top 1 intruder dimension in every weight matrix in the model and scale it by a multiplicative constant. For example, if the top 1 intruder dimension is at index $i$, we have $W = W_0 + (\Delta W - u_i * \sigma_i * v_i^T) + \lambda(u_i * \sigma_i * v_i^T)$. $\lambda = 1$ is no change. We find that reducing the scale of the top intruder dimension, while only slightly impacting the test accuracy, leads to a large drop in pre-training loss. For example, in the rank 8 case, simply deleting the top intruder dimension in each matrix leads to a 26.1% drop (lower is better) in loss with only a 5.9% drop in test performance. Note that test accuracy doesn't drop to baseline with $\lambda = 0$ because we haven't removed the entire update but instead only the top intruder dimension (if it exists) in the weight matrix. This suggests that intruder dimensions are responsible for most of the drop in OOD performance and only a small portion of the total learning the model undergoes. Our baseline, which instead removes a neighboring singular vector to the intruder dimension, degrades immediately, suggesting that singular vectors that are not intruder dimensions are essential to model performance.

1242
1243
1244
1245
1246
1247
1248
1249
1250
1251
1252
1253
1254
1255
1256
1257
1258
1259
1260
1261
1262
1263
1264
1265
1266
1267
1268
1269
1270
1271
1272
1273
1274
1275
1276
1277
1278
1279
1280
1281
1282
1283
1284
1285
1286
1287
1288
1289
1290
1291
1292
1293
1294
1295

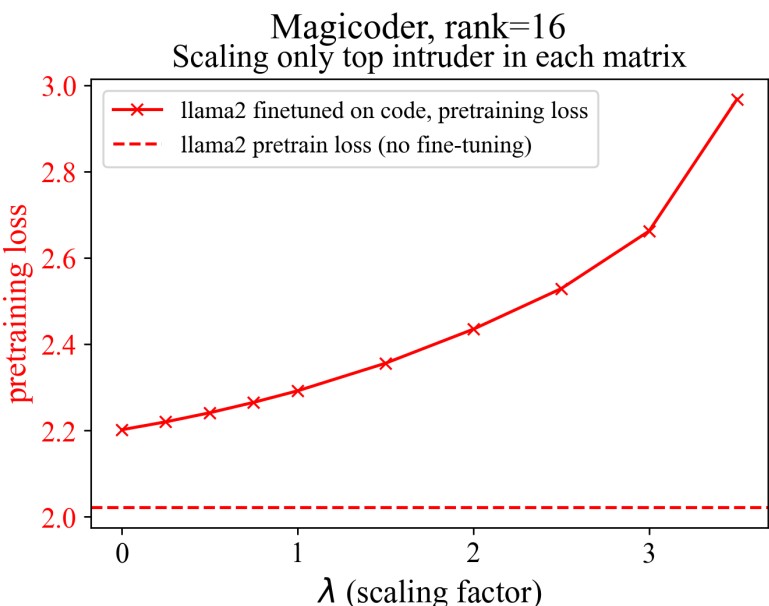

Figure 23: **Scaling LLaMA2 Intruder Dimensions.** To confirm our findings from Fig. 22, we repeat our methodology but now on a LLaMA2 model fine-tuned on code with a rank 16 LoRA. We find that scaling down the top intruder dimension in each matrix leads to lower pre-training loss, while scaling up the top intruder dimension leads to higher loss pre-training loss.