# OpenReview forum: "LoRA vs Full Fine-tuning: An Illusion of Equivalence"
_ICLR.cc/2025/Conference — Submitted to ICLR 2025_

### Official Review · Reviewer_UCu1 · 2024-10-29

**Soundness:** 1
**Presentation:** 2
**Contribution:** 1
**Rating:** 3
**Confidence:** 4

**Summary:**

This paper empirically examines behavioral differences between LoRA and full fine-tuning, revealing that LoRA introduces *intruder dimensions* — singular vectors with low cosine similarity to the top $k$ singular vectors in the original weight matrix. Centered on this observation, the paper lists approximately ten empirical differences between LoRA and full fine-tuning.

**Strengths:**

The paper presents an interesting observation on behavioral differences between LoRA and full fine-tuning, followed by a series of experiments exploring various aspects of these observations.

**Weaknesses:**

This paper should be rejected because:

1. While the observation is interesting, the paper does not establish why the intruder dimension is significant. Counterexamples can be easily constructed to demonstrate that the intruder dimension may not materially impact the model. For instance, consider a 3-D space where the original weight matrix has two singular vectors, (1, 0, 0) and (0, 1, 0). Now consider a LoRA fine-tuned weight matrix with singular vectors (1, 1, 0) and (1, -1, 0). Both vectors are intruder dimensions, yet they span the same subspace as the original vectors and may not affect the fine-tuning process positively or negatively.

2. The paper points out some weaknesses of LoRA (e.g., full fine-tuning achieves a higher effective rank than LoRA), implying that the intruder dimension may be the root cause. However, the paper does not establish the connection between intruder dimension and effective rank, either theoretically or empirically.

3. The paper does not explore initialization, a key difference between LoRA (in which either $A$ or $B$ is randomly initialized) and full fine-tuning (which requires no initialization). Initialization could directly influence the intruder dimension, raising doubts about whether the empirical results stem from favorable initializations that support the paper's claims.

4. The experiments are narrowly focused, examining only cases where LoRA and full fine-tuning have the same accuracy, only sequence tasks, and only a limited range of hyperparameters, suggesting that the findings may be limited to a small niche rather than universally applicable.

Additionally, the paper has several confusing parts:

- The algorithm in Figure 4 mandates that num_intruders $\leq k$, yet Figure 5 shows num_intruders at O(100) while $k = 10$ .
- The results in Figure 8 do not support the claim; in some cases, full fine-tuning has lower accuracy than LoRA, and in others, full fine-tuning's accuracy is even lower than LoRA with a lower rank.
- The results in Figure 9 suggest there is always a LoRA rank where the pseudo-loss is lower than that of full fine-tuning. Does this mean that LoRA could be superior to full fine-tuning if the LoRA rank $r$ is correctly tuned?
- In some experiments, the paper uses $\alpha$ values that are not multiples of $r$, deviating from common LoRA practices. While these setups yielded interesting results, the paper may benefit from experimenting with different $\alpha / r$ ratios to provide insights into LoRA’s behavior in common configurations.

**Questions:**

1. Why should readers care about *intruder dimensions*?
2. What is the theoretical or empirical connection between intruder dimensions and effective rank?
3. How might initialization affect intruder dimensions?
4. Do we observe similar behaviors in more general cases, such as when LoRA and full fine-tuning differ in accuracy, for non-sequence tasks, or with different hyperparameters?
5. How are there O(100) intruder dimensions if Figure 4’s algorithm limits the number to $k=10$?
6. Does Figure 9 suggest that, with correctly tuned $r$, LoRA could achieve lower pseudo-loss than full fine-tuning?
7. How would behavior change if $\alpha / r$ is kept constant at a value other than 2?

---

> ### Author Response · Authors · 2024-11-21
> **UCu1 (1/2)**
>
> We thank you for your review.
>
> Weaknesses:
>
> > While the observation is interesting, the paper does not establish why the intruder dimension is significant.
>
> Intruder dimensions appear as a result of the diminished expressivity of the LoRA update and cause a characteristic decline in the model's behavior out-of-distribution. To clarify the significance of the intruder dimensions, we show that intruder dimensions lead to worse OOD performance with a new experiment, described in the general response.
> Intruder dimensions are significant because they cause a clear structural difference in the weight matrices between LoRA and full fine-tuning and show a clear divergence in mechanism for learning between the two methods. While full fine-tuning makes small changes to the singular vectors and values of the pre-trained weights, LoRA introduces these new singular vectors (intruder dimensions) that are very unlike the pre-trained singular values.
>
> > The paper does not establish the connection between intruder dimension and effective rank, either theoretically or empirically.
>
> Our study of effective rank is intended as an interesting aside and is not important or essential to our findings and does not affect them. However, arguments can be made to justify their connection: if you have high ranking intruder dimensions, this means you have new singular values that are large. This in turn makes the distribution sharper and therefore will decrease the effective rank.
>
> > The paper does not explore initialization.
>
> When initializing our LoRA modules, we use the standard and widely used initialization described in [1]. Per your advice, we measure the impact of random seed on the number of intruder dimensions by initializing and fine-tuning with 5 different seeds. We find that there is almost no difference across these runs. (See appendix for this plot).
>
> > The experiments are narrowly focused, examining only cases where LoRA and full fine-tuning have the same accuracy.
>
> This is intentional and is what makes the finding surprising. We intentionally examine cases where LoRA and full fine-tuning have the same accuracy in order to fairly examine if despite their equivalence in-distribution, do the methods utilize different parts of the parameter space? If we did not do this, conclusions would be harder to generalize because of differing degrees of fit between the two methods.
>
> However, we additionally conduct new experiments to help address this concern. In them, we examine the impact of learning rate and number of epochs on the number of intruder dimensions, test performance, and pre-training loss. This enables us to cover a wider range of scenarios for these models. We further describe these experiments in the main text and report them in Figure 20 (Page 22) of the updated pdf. We hope this strengthens our conclusions.
>
> Clarifying Points of Confusion:
> > num_intruders O(100) while k=10, even though it is mandated that num_intruders<=k.
>
> In our plots, we report values across all weight matrices in the model, while k=10 is enforced on each individual weight matrix. Since RoBERTa has 72 linear layers (excluding embedding), we have an upper limit of 72*10 possible intruder dimensions when k=10. We apologize for this confusion. We have modified the text in the paper (line 291-293) to reflect this change.
>
> > The results in Figure 8 do not support the claim; in some cases full fine-tuning has lower accuracy than LoRA, and in others, full fine-tuning’s accuracy is even lower than LoRA with low rank.
>
> In Figure 8, we do not make the claim that full fine-tuning has higher accuracy than LoRA. We are instead focused on the *forgetting* that these models exhibit. As we can see on MNLI, QQP, SST2, even though all methods train to roughly similar accuracy, they exhibit clearly different forgetting curves (the white area), with lower ranks of LoRA, which have many intruder dimensions, forgetting much more than both full fine-tuning and high ranks of LoRA. On the other plots that you are pointing out, the models *do not train to the same initial accuracy* (location of black vertical line), so forgetting claims based on these findings are invalid on these datasets. Again, we are interested in *forgetting* and in no way are claiming that full fine-tuning should always have higher accuracy than LoRA. Also, please note that the gray area represents performance before the model has been trained on the particular task. Therefore, the gray region is irrelevant to us.

---

> > ### Comment · Reviewer_UCu1 · 2024-11-25
> >
> > Thanks for including Figures 21 and 22 in the revision. It's good to know that random initialization did not make any difference and that intruder dimensions behave differently from pre-trained singular vectors. Also, thank you for clarifying that the values in Figure 5 represent a sum over multiple layers.
> >
> > However, I'm afraid my main concern still remains: why is the concept of intruder dimensions significant enough to warrant its introduction as a new concept? Some of my earlier questions, such as how intruder dimensions relate to subspace and effective rank, were intended to explore their connection to established concepts, which might provide a better answer to this concern. My threshold for introducing a new concept (which, of course, is up for debate) is that the following criteria should ideally be met:
> >
> > 1. It can explain phenomena that existing, well-established concepts already explain.
> > 2. Additionally, it can explain phenomena that existing, well-established concepts cannot.
> >
> > Do you have specific comments on subspace in this context?
> >
> > Additionally, I still have follow-up questions on the following two points:
> >
> > > The experiments are narrowly focused, examining only cases where LoRA and full fine-tuning have the same accuracy.
> >
> > My concern is whether the observations apply only to narrow conditions and may not generalize to other scenarios, such as when LoRA and full fine-tuning achieve different accuracies (whether better or worse).
> >
> > > The results in Figure 8 do not support the claim; in some cases, full fine-tuning has lower accuracy than LoRA, and in others, full fine-tuning’s accuracy is even lower than LoRA with a lower rank.
> >
> > The inconsistency I noted was not in the gray area but in the white area of Figure 8. I couldn't identify a consistent trend between the accuracy of LoRA and full fine-tuning.

---

> ### Author Response · Authors · 2024-11-21
> **UCu1 (2/2)**
>
> > Do the results in Figure 9 suggest that LoRA could be superior to full fine-tuning if LoRA rank r is correctly tuned?
>
> Correct. The results in Figure 9 suggest that at the same accuracy level, certain ranks of LoRA may forget less than full fine-tuning. Therefore, it is indeed the case that LoRA may be better than full fine-tuning if the rank is correctly tuned.
>
> > The paper uses alpha values that are not multiples of r, deviating from common LoRA practices…The paper may benefit from experimenting with different alpha/r ratios.
>
> We study two standard alpha settings. The alpha=8, which was used in the original LoRA paper [[1]], and the currently recommended value of alpha=2*r [[2]].  Our main results are reported with alpha=2*r, which is the common LoRA practice as you stated. Ablating the alpha settings and finding their impact is an interesting topic for future work.
>
> [1]: https://arxiv.org/abs/2106.09685
>
> [2]: https://arxiv.org/abs/2405.09673

---

> > ### Comment · Reviewer_UCu1 · 2024-11-25
> >
> > > LoRA may be better than full fine-tuning if the rank is correctly tuned.
> >
> > Agreed! So rather than portraying intruder dimensions as purely detrimental, perhaps there should be work exploring why they might actually enable LoRA to outperform full fine-tuning in certain scenarios. :)
> >
> > > Ablating the alpha settings and finding their impact is an interesting topic for future work.
> >
> > My intent was to ensure that the observed phenomena aren't isolated to a specific setup but hold water across different configurations.

---

> > ### Author Response · Authors · 2024-12-02
> >
> > Since the reviewing window is closing soon, please let us know if there are any more concerns you have with our work that you would like us to address. If you have no more concerns, we ask that you raise your score to reflect this.

---

> ### Author Response · Authors · 2024-11-26
>
> **(1/2)**
> > I'm afraid my main concern still remains: why is the concept of intruder dimensions significant enough to warrant its introduction as a new concept?
>
> Your concern about the need to introduce a new concept is valid and shared by another reviewer. Here is a similar response to what we gave them:
>
> The reason we need a new measure is that effective rank or other subspace measures are general measurements, but do not pin-point where the change is. Our measure is instead localized---which in this case is critical. This allows us to identify and point to the exact singular vector that is different across the pre-trained and fine-tuned weight matrices. In fact, our experiment in Figure 22, in which we exclusively scale the intruder dimensions and show that they are responsible for worse OOD performance, would not be possible with a general measure. These means that intruder dimensions explain the worse OOD performance while other existing, well-established concepts like effective rank cannot because they are general and not specific.
>
> Measuring effective rank or something similar cannot capture the "intruder" dimensions because they are measurements that capture the properties of the entire matrix and do not pin-point the precise difference between the methods--i.e, the shifting of the pre-trained singular vectors and the "new" intruder dimensions. Please let us know if you have any specific measures in mind that would allow us to capture the nature of this update precisely without introducing a new concept.
>
> > The experiments are narrowly focused, examining only cases where LoRA and full fine-tuning have the same accuracy.
>
> To re-emphasize, we have studied LLaMA-7b/LLaMA2-7b models that were fine-tuned on tasks on which LoRA and full fine-tuning do not reach same performance and still find that lower ranks of LoRA have intruder dimensions, showing that our conclusions hold. Furthermore, as we stated in the general response, we are using standard tasks to benchmark LoRA and full fine-tuning([1], [2], [3]). None of these works used tasks that LoRA and full fine-tuning diverge on to benchmark them. To quote our general response, "We point out that more computational resources are required to fine-tune models for code generation or other long form tasks and is a limiting factor for us. While we do have intruder dimension measurements on models for code generation and other long-form tasks, we do not have pre-training loss experiments for these because it is widely found that LoRA doesn't match the performance of full fine-tuning on tasks like code generation([5]) and long-form text([6]) and therefore any OOD analysis would diverge from our premise of 'measuring OOD performance given the same IID performance'."
>
> > The inconsistency I noted was not in the gray area but in the white area of Figure 8. I couldn't identify a consistent trend between the accuracy of LoRA and full fine-tuning.
>
> 1. First, we can see that rank 1(yellow) always has lower accuracy than full fine-tuning(blue).
> 2. In this setting we are actually interested in *forgetting* during continual learning. When different methods train to similar accuracy, we can examine if a certain method forgets more or less than another. In these plots, we see that **low ranks of LoRA,** which are models that contain intruder dimensions, have worse forgetting in comparison to **both high ranks of LoRA and full fine-tuning**.
> 3. For example, looking at the top left plot for MNLI, the vertical black line indicates when we train on the task. we see that all methods reach a similar accuracy. Then, going right, we see that the yellow(r=1) and orange(r=8) lines show that they forget more in comparison to the high ranks of LoRA and full fine-tuning. Note this pattern is also clear for QQP and SST-2: all methods train to similar accuracy, but the yellow and orange lines clearly decrease more than the others.
> 4. Please note that it is true that high ranks of LoRA(r=64, r=768) sometimes forget less in comparison to full fine-tuning. These models do not have intruder dimensions. We are using this experiment to further justify that models with intruder dimensions forget more.

---

> > ### Comment · Reviewer_UCu1 · 2024-11-26
> >
> > > Why is the concept of intruder dimensions significant enough to warrant its introduction as a new concept?
> >
> > I would be more readily convinced if I saw a rigorous discussion on why subspace or effective rank cannot explain what intruder dimensions explain. As I mentioned earlier, exploring the connections to established concepts might provide a more satisfactory answer to this concern. While it may be true that the experiment in Figure 22 cannot be easily conducted using subspace or effective rank, it is not rigorous to claim that this alone justifies introducing intruder dimensions. Alternatively, there might be ways to tailor the experiments for subspace or effective rank. One possibility could involve scaling the singular vectors that are cosine-similar to the intruder dimensions, instead of scaling the singular vectors from a similar location. And this is just one setup among many possible approaches.
> >
> > Additionally, I want to re-emphasize that I am less interested in establishing a new concept. I would like to see to what extent existing well-established concepts can and cannot (both aspects are crucial) explain the phenomena observed in this paper.
> >
> > > The experiments are narrowly focused, examining only cases where LoRA and full fine-tuning have the same accuracy.
> >
> > My primary question here is whether intruder dimensions have explanatory power only in cases where LoRA and full fine-tuning achieve the same accuracy. This would help determine if intruder dimensions represent a narrowly scoped phenomenon or a more generic one. I think they might have broader explanatory power, but it is up to the authors to rigorously establish this.
> >
> > > The inconsistency I noted was not in the gray area but in the white area of Figure 8.
> >
> > The most problematic subplot is SIQA, where lower-ranked LoRA performs better in the white area, while full fine-tuning and higher-ranked LoRA do not. The MNLI subplot is also problematic, as full fine-tuning consistently underperforms compared to higher-ranked LoRA.

---

> ### Author Response · Authors · 2024-11-26
>
> **(2/2)**
> > Rather than portraying intruder dimensions as purely detrimental, perhaps there should be work exploring why they might actually enable LoRA to outperform full fine-tuning in certain scenarios.
>
> This is not the stance that we take. As shown in Fig. 22, intruder dimensions are causally responsible for worse OOD performance. Furthermore, higher ranks of LoRA, which perform better, have *fewer intruder dimensions*. Because of this, we do not believe that intruder dimensions are responsible for LoRA outperforming full fine-tuning.
>
> > Ablating the alpha settings and finding their impact is an interesting topic for future work.
>
> Our results are reported with the two dominantly used alpha settings: $\alpha=8$ from the original LoRA paper[1], and $\alpha=2r$, which is now widely accepted as the best setting to use[4,5]. Trying new alpha values would require rerunning every single experiment, **including fine-tuning**. Depending on hardware availability, rerunning these experiments could take many days to run for just a single alpha value. Because of the cost of running these experiments and our use of the common and widely accepted alpha settings, we do not think it is a reasonable request to rerun using many other alpha settings in order to show that our findings hold.
>
> citations:
>
> [1]: LoRA: https://arxiv.org/abs/2106.09685
>
> [2]: AdaLoRA: https://arxiv.org/abs/2303.10512
>
> [3]: VeRA: https://arxiv.org/abs/2310.11454
>
> [4]: rsLoRA: https://arxiv.org/abs/2312.03732
>
> [5]: LoRA Learns Less: https://arxiv.org/abs/2405.09673
>
> [6]: Tulu 2: https://arxiv.org/abs/2311.10702

---

> > ### Comment · Reviewer_UCu1 · 2024-11-26
> >
> > > Because of this, we do not believe that intruder dimensions are responsible for LoRA outperforming full fine-tuning.
> >
> > Good to know. I believe this is an important message that the authors want to convey to the audience.
> >
> > > Ablating the alpha settings and finding their impact is an interesting topic for future work.
> >
> > They are the settings commonly used for LoRA fine-tuning do not necessarily mean they are the only configurations this research should explore. Once again, I think intruder dimensions might have broader explanatory power, but it is up to the authors to rigorously establish this.

---

> > > ### Author Response · Authors · 2024-12-02
> > >
> > > > I would be more readily convinced if I saw a rigorous discussion on why subspace or effective rank cannot explain what intruder dimensions explain.
> > >
> > > What does the reviewer mean by "subspace"? We are confused what they mean by this. See below for why effective rank cannot explain what intruder dimensions explain.
> > >
> > > > One possibility could involve scaling the singular vectors that are cosine-similar to the intruder dimensions.
> > >
> > > We are concerned that the reviewer does not understand the definition of intruder dimensions. By definition, intruder dimensions have *low* cosine similarity to the pre-trained singular vectors. Therefore, this measurement would be misguided. If the reviewer instead is talking about the fine-tuned singular vectors, we point out that in the SVD singular vectors are all *orthogonal* to each other.
> > >
> > > > I would like to see to what extent existing well-established concepts can and cannot (both aspects are crucial) explain the phenomena observed in this paper.
> > >
> > > To appease the reviewer, we measure the effective rank of our trained models and correlate it with forgetting on the pre-training distribution. When we do this, we get p-value$>0.3$, meaning that there is not a statistically significant relationship between effective rank and forgetting. In contrast, there is a clear correlation between intruder dimensions and forgetting($\rho=0.944$, p-value$<0.001$), as we described in Fig. 20.
> > >
> > > > My primary question here is whether intruder dimensions have explanatory power only in cases where LoRA and full fine-tuning achieve the same accuracy.
> > >
> > > Based on your suggestion, we replicate our methodology from Fig. 22(where we scale intruder dimensions) on LLaMA2 and report our findings in Fig. 23 in the updated pdf. We find that **intruder dimensions still have explanatory power when full fine-tuning and LoRA have different accuracy:** scaling down intruder dimensions causes to less forgetting, and scaling up intruder dimensions causes more forgetting.
> > >
> > > > The most problematic subplot is SIQA, where lower-ranked LoRA performs better in the white area, while full fine-tuning and higher-ranked LoRA do not.
> > >
> > > Because the models did not initially train to similar initial performance, we are unable to make comparisons of forgetting on this plot. To reiterate, we are focused on *forgetting*, not in distribution test accuracy. Therefore, this plot is inconclusive but not harmful to our claims. We presented this plot for completeness.
> > >
> > > > The MNLI subplot is also problematic, as full fine-tuning consistently underperforms compared to higher-ranked LoRA.
> > >
> > > We ask the reviewer to take a closer look at our previous response. In it, we said:
> > >
> > > "Please note that it is true that high ranks of LoRA(r=64, r=768) sometimes forget less in comparison to full fine-tuning. These models do not have intruder dimensions. We are using this experiment to further justify that models with intruder dimensions forget more."
> > >
> > > To reiterate, we make no claim that full fine-tuning must forget less in comparison to high rank LoRAs that have no intruder dimensions. Therefore, this plot **is not problematic**, because full fine-tuning is outperforming LoRA models that have many intruder dimensions.

---

### Official Review · Reviewer_UXpu · 2024-10-31

**Soundness:** 3
**Presentation:** 3
**Contribution:** 2
**Rating:** 5
**Confidence:** 4

**Summary:**

This paper finds that full fine-tuning and LoRA yield weight matrices with distinctly different structures in their singular value decompositions, even though their downstream task performance is similar. The authors introduce *Intruder Dimensions* to analyze these differences, offering valuable insights from extensive experiments on LoRA training with varying ranks and $\alpha$ values. Additionally, they explore how *Intruder Dimensions* impact the model’s expressive power and generalization.

**Strengths:**

- The writing is clear, and the proposed method is well-motivated and sound.

- The authors introduce *Intruder Dimensions* to analyze the differences between LoRA and full fine-tuning, conducting extensive experiments to investigate this phenomenon. They present several findings regarding the generalization of LoRA fine-tuned models and offer suggestions, such as scaling $\alpha$ with rank and freezing A, to reduce *Intruder Dimensions*.

**Weaknesses:**

The main weakness of this paper is that, as an analysis-focused study, the exploration of *Intruder Dimensions* is somewhat **straightforward**. However, the analysis lacks depth; for example, in Section 5, the proposed hypotheses are not thoroughly explored theoretically. While the authors present several findings, they do not provide additional constructive improvement measures, limiting the paper’s practical contribution to the community.

**Questions:**

- When comparing LoRA to full fine-tuning, I suggest conducting additional experiments to analyze the impact of hyper-parameters. Specifically, for LoRA, variations in learning rate and total training steps can influence the final parameters.
  - For instance, in Figure 7, adding a plot showing accuracy over training steps would be helpful.

- I recommend including some analysis of *Intruder Dimensions* for more LoRA variants, such as VeRA [r1], LoRA+ [r2], and FourierFT [r3], which would enhance the significance of the work.



  [r1] Kopiczko, Dawid Jan, Tijmen Blankevoort, and Yuki M. Asano. "VeRA: Vector-based Random Matrix Adaptation." ICLR 2024.

  [r2] Hayou, Soufiane, Nikhil Ghosh, and Bin Yu. "LoRA+: Efficient Low Rank Adaptation of Large Models." ICML 2024.

  [r3] Gao, Ziqi, et al. "Parameter-Efficient Fine-Tuning with Discrete Fourier Transform." ICML 2024.

---

> ### Author Response · Authors · 2024-11-21
> **UXpu**
>
> We thank you for your review.
>
> Weaknesses:
> > the analysis lacks depth; for example, in Section 5, the proposed hypotheses are not thoroughly explored theoretically.
>
> Our paper is primarily empirical and experimental. It is important to note that intruder dimensions are an empirical observation. In this paper, we provide several mathemstical justifications for their occurrence. We showed that adding the outer product of a random vector to a weight matrix introduces an intruder dimension. This is analogous to the matrix product of BA in LoRA when rank is 1, and implies that intruder dimensions occur because the B and A vectors are uncorrelated to the columns/rows of the weight matrix W_0. Further, we showed empirically that just training the B matrix, while freezing the A matrix with all singular values of 1, eliminates the amplification of the singular values because of the matrix product and therefore reduces the number of high ranking intruder dimensions.
>
> > While the authors present several findings, they do not provide additional constructive improvement measures.
>
> We provide several recommendations to improve LoRA:
> 1. We show the importance of tuning rank and suggest that even if you can get the same test accuracy with a lower rank, using a rank of 64 is still preferable because of less forgetting of previously learned information. These results emphasize that the selection of rank is still important, even with the same test accuracy.
> 2. We show that the selection of alpha is important for the performance of LoRA and can lead to poor models, even when test accuracy is similar across values of alpha. We find that setting alpha=2*r is indeed optimal, confirming others([1],[2]) and setting alpha to a constant value for high ranks can lead to poor performing models (unfortunately, many papers still do this).
> 3. Based on our new experiment in response to [UXpu, UCu1] and detailed in Figure 20 (Page 22), We show that using a lower learning rate decreases the number of intruder dimensions and has better OOD performance. Therefore, we advise the use of the lowest learning rate that can be used to fit to the fine-tuning task.
>
> Questions:
> > I suggest conducting additional experiments to analyze the impact of hyperparameters. Specifically, for LoRA, variations in learning rate and total training steps can influence the final parameters.
>
> Per your feedback, we conduct this analysis and report this in Figure 20 (Page 22) of the updated PDF. We find that learning rate does indeed play an important role in what we observe. Please see the general response for more details.
>
> > In Figure 7, add a plot showing accuracy.
>
> Following your request, we have added a plot showing accuracy over training steps to Figure 7  (Page 7). We hope this is helpful for understanding our findings. (See new submission doc for new figure.)
>
>
> >I recommend including some analysis of Intruder Dimensions for more LoRA variants, such as VeRA, LoRA+, and FourierFT.
>
> We thank you for your recommendation to enhance this work. We have run intruder dimension experiments on these three different methods to see the differences across these methods and report them in our overall response.
>
> Citations:
>
> [1] https://arxiv.org/abs/2405.09673
>
> [2] https://arxiv.org/abs/2312.03732

---

> > ### Comment · Reviewer_UXpu · 2024-11-21
> >
> > Thank you for your response! Most of my concerns have been addressed. There are still a few points I’d like to seek clarification on:
> >
> > - In Figure 20, we observe that intruder dimensions are significantly reduced. Could this imply that at lower learning rates (e.g., 1e-5), LoRA approaches equivalence to full fine-tuning?
> > - You mentioned that using higher-rank LoRA is still preferable due to less forgetting of previously learned information. However, the primary purpose of LoRA is to use fewer parameters and adapt quickly to downstream tasks. To handle more tasks, we could also employ multiple low-rank LoRA models. In that case, would MoE (Mixture of Experts) be a more suitable approach?

---

> ### Author Response · Authors · 2024-11-21
>
> We are glad your concerns were addressed! Your review was very helpful for improving this work.
>
> Responses:
>
> - Yes, this could imply that with very low learning rates, the number of intruder dimensions may go to zero. However, the concern with using even lower learning rate will be it takes longer to converge. The default learning rate for LoRA in the PEFT library is 1e-4[1], and established papers [2,3] use a learning rate of 5e-4 for our model-dataset combo. We use the learning rate you mention to train a new model and compare it to 1e-4 (the default learning rate in the PEFT library [1]). We find that using 1e-5 is unable to reach the peak performance of 1e-4, even after training for **5x longer**. We have added this to the list of recommendations since this is a setting no-one currently uses and is atypical.
>
> - When trying to handle many different tasks well with one model, It is indeed a good suggestion to use a Mixture of Experts(MoE) model instead of updating a single model. While our paper is highlighting this forgetting problem that was previously not known, we will include your recommendation of employing multiple low-rank LoRA adapters to help alleviate this forgetting in our discussion section. However, it is important to note that this remedy will add an increasing number of parameters for each task since we need a new adapter per task.
>
> Citations:
>
> [1]: PEFT: https://huggingface.co/docs/diffusers/training/lora
>
> [2] LoRA paper: https://arxiv.org/abs/2106.09685
>
> [3] AdaLoRA paper: https://arxiv.org/abs/2303.10512

---

> > ### Author Response · Authors · 2024-12-02
> >
> > Since the reviewing window is closing soon, please let us know if there are any more concerns you have with our work that you would like us to address. If you have no more concerns, we ask that you raise your score to reflect this.

---

### Official Review · Reviewer_w6KM · 2024-11-02

**Soundness:** 2
**Presentation:** 4
**Contribution:** 2
**Rating:** 6
**Confidence:** 3

**Summary:**

This paper compares LoRA and full fine-tuning in adapting pre-trained large language models to downstream tasks. The study reveals that LoRA introduces "intruder dimensions" in the weight matrices, which do not appear in fully fine-tuned models. The paper provides empirical evidence for that (1) LoRA models with intruder dimensions tend to forget more of the pre-training distribution and adapt less robustly to multiple tasks sequentially compared to full fine-tuning; (2) Higher-rank LoRA models, which more closely resemble full fine-tuning, perform better in terms of generalization and robustness; (3) Intruder dimensions negatively impact model performance outside the fine-tuning task distribution.

**Strengths:**

1. The paper provides a novel insight into the differences between LoRA and full fine-tuning by analyzing the spectral properties of the weight matrices. The concept of "intruder dimensions" offers a new perspective on understanding these methods.
2. The paper not only identifies the presence of intruder dimensions but also analyzes their impact on model behavior, providing evidence for their detrimental effects.

**Weaknesses:**

1.While the paper provides empirical evidence for the existence and impact of intruder dimensions, the theoretical explanation for why they appear in LoRA models is still lacking.
2.The experiments are mainly conducted on sequence classification tasks and RoBERTa. The experiments on generative tasks seem insufficient and only preliminary attempts have been made, leading to doubts about whether the conclusions of this paper can be generalized to autoregressive models.
3.The authors focus on the original LoRA setting, while ignoring the other LoRA fine-tuning technology.

**Questions:**

1. Line 173: If the differences in training tasks lead to completely opposite conclusions with (Biderman et al., 2024), then the theoretical analysis of LoRA in this paper may not be comprehensive. Therefore, I am curious whether this phenomenon is caused by the degree of fitting or other parameters.
2. How do LoRA and full fine-tuning perform on more complex tasks like code generation or long-form text generation and autoregressive LLMs? Do the findings on sequence classification tasks generalize to these settings?
3. I am curious whether the phenomenon studied in this paper can be generalized to some recent LoRA fine-tuning techniques, such as QLoRA, Pissa, LoRA+, etc. If these commonly used training techniques do not encounter "intruder dimensions", the significance of this work will be greatly reduced.

---

> ### Author Response · Authors · 2024-11-21
> **w6KM**
>
> We thank you for your review.
>
> Weaknesses:
>
> > While the paper provides empirical evidence for the existence and impact of intruder dimensions, the theoretical explanation for why they appear in LoRA models is still lacking.
>
> Intruder dimensions are an empirical observation of fine-tuning, and we provide a few mathematical justifications for their occurrence. We showed that adding the outer product of a random vector to a weight matrix introduces an intruder dimension. This is analogous to the matrix product of BA in LoRA when rank is 1, and implies that intruder dimensions occur because the B and A vectors are uncorrelated to the columns/rows of the weight matrix W_0. Further, we showed empirically that just training the B matrix, while freezing the A matrix with all singular values of 1, eliminates the amplification of the singular values because of the matrix product and therefore reduces the number of high ranking intruder dimensions.
>
> Empirically, the appearence of intruder dimensions is a function of the data distribution, rank, and learning rate. An exact theoretical understanding of why intruder dimensions appear, and why their appearence is affected by different factors is outside the scope of the current paper, will require substantial theoretical advancement and would be an important direction for future research.
>
> > The experiments on generative tasks seem insufficient and only preliminary attempts have been made, leading to doubts about whether the conclusions of this paper can be generalized to autoregressive models.
>
> We analyzed the intruder dimension of 3 different generative tasks in this work: instruction following (Fig. 5b), code generation (Fig. 5c), and math (Fig. 5d). All of these experiments confirm our findings on RoBERTa models: using LoRA with lower ranks consistently leads to the introduction of intruder dimensions, while using higher ranks of LoRA leads to fewer.
>
> > The authors focus on the original LoRA setting, while ignoring other LoRA fine-tuning technology.
>
> In this work, we focused on an in depth study of the original LoRA setting and left other variants for future work. However, per your advice, we analyze the intruder dimensions of PiSSA and LoRA+, as mentioned in the general response.
>
> Questions:
>
> > If the differences in training tasks lead to completely opposite conclusions with (Biderman et al., 2024), then the theoretical analysis of LoRA in this paper may not be comprehensive. Therefore, I am curious whether this phenomenon is caused by the degree of fitting or other parameters.
>
>
> Biderman et al., 2024 evaluated models where LoRA did worse than full fine-tuning, whereas we primarily evaluate settings where LoRA matches full-finetuning performance. While Biderman et al., 2024 found the LoRA forgot less when it fit worse to the test data, we found that, when trained to similar fits on the test data, certain ranks of LoRA forget more. These difference in conclusions are due to the differences in the setup between our paper and Biderman et al., 2024.
>
> > How do LoRA and full fine-tuning perform on more complex tasks like code generation or long-form text generation and autoregressive LLMs? Do the findings on sequence classification tasks generalize to these settings?
>
> The current literature is mixed with regard to the ability of LoRA to match full fine-tuning on more complex tasks. For example, Dettmers et al. [[1]], which proposed QLoRA, found that they could reach SOTA chatbot performance when doing QLoRA with instruction following, outperforming fully fine-tuned models. However, Biderman et al. [[2]] found that there was a gap in performance between LoRA and full fine-tuning on code generation and math.
> Our findings regarding the presence of intruder dimensions do generalize to these settings, as shown in Fig 5b, 5c, & 5d.
>
> > I am curious whether the phenomenon studied in this paper can be generalized to some recent LoRA techniques, such as QLoRA, Pissa, LoRA+, etc.
>
> We thank you for this suggestion to improve this work. In the current manuscript, Fig 5b contains analysis of a QLoRA model and shows that it has a similar number of intruder dimensions, across different values of epsilon, as regular LoRA. Per your suggestion, we have analyzed the intruder dimensions of PiSSA and LoRA+ to see the differences across these methods and report them in our overall response.
>
> [1]: https://arxiv.org/abs/2305.14314
>
> [2]: https://arxiv.org/abs/2405.09673

---

> > ### Author Response · Authors · 2024-12-02
> >
> > Since the reviewing window is closing soon, please let us know if there are any more concerns you have with our work that you would like us to address. If you have no more concerns, we ask that you raise your score to reflect this.

---

### Official Review · Reviewer_NsNJ · 2024-11-05

**Soundness:** 2
**Presentation:** 3
**Contribution:** 2
**Rating:** 5
**Confidence:** 4

**Summary:**

In this paper, to analyze the difference between LoRA fine-tuning and full-weight fine-tuning, the authors propose a new concept, intruder dimension, that measures the similarity of the column basis of singular-value decomposed matrices between the pre-trained weights and fine-tuned weights. Based on the concept of intruder dimension, the authors empirically analyze the difference by visualizing different models' performance and their number of intruder dimensions, evaluated on multiple datasets. The authors made several observations about LoRA's generalization ability to out-of-distribution datasets and drew several conclusions based on the observations. Finally, the authors compare the performance of continual learning abilities of different ranks of LoRA.

**Strengths:**

1. This paper provides a new concept for analyzing LoRA's different behavior.
2. This paper has abundant empirical evidence showing the difference between the LoRA and full-weight fine-tune, based on the proposed concept.
3. The paper is organized in a straightforward way and can be easily comprehended.

**Weaknesses:**

- **The proposed concept "intruder dimension" is limited in its scope and cannot be readily extended to other scenarios.**
  + According to Definition 1, the intruder dimension depends on the setting of the threshold $\epsilon$, which varies with different number of dimension (footnote 1, page 4).
  + The concept of intruder dimension considers only the rotational effect of a linear mapping, and it holds only the weight increment $\Delta W$ is small compared to the pre-trained weights. To see this, consider a pre-trained square matrix $W\_0=U\_0 \Sigma\_0 V\_0^T$ (it can also be extended to non-square matrix), and do the SVD to the weight increment as following: $\Delta W = U \Sigma V^T = U\_0 Q \Sigma Q^T V\_0^T$, where $Q$ is a orthonormal (rotation) matrix. Then the fine-tuned matrix can be represented by: $W\_0 + \Delta W = U\_0 (\Sigma\_0 + Q \Sigma Q^T) V\_0^T$. The transformation in between the orthogonal basis $(\Sigma\_0 + Q \Sigma Q^T)$ is the one we actually want to analyze. Doing a further SVD to this will give us $(\Sigma\_0 + Q \Sigma Q^T) = P \Sigma^\prime P^T$. The similarity matrix on which the **intruder dimension** then can be calculated by the matrix product $(U\_0P)^T U\_0=P^T$. It can be seen as the effective rotational transformation learned by the $\Delta W$ and no consideration is paid to the scaling effect, i.e., $\Sigma^\prime$.
  + Following the second bullet, I don't see the necessity of proposing a whole new concept for analyzing this rotational transformation $P$, as we have well-established inner-product for two different orthonormal matrices, e.g., Frobenius inner-product $<P, Q>=tr(P^TQ)$. And in this case, $tr(P)$. Could you please explain why do we need to define such a new concept?
- I'm confused about the conclusion of this paper: **is intruder dimension the reason for worse OOD generalization or the consequence of LoRA?** It would be good to confirm the answer to this question. Note it brings a huge difference:
  + If it's the cause then ideally we can actively mitigate this issue by limiting the intruder dimension, i.e., restricting the rotation transformation of LoRA and we should expect a gain in performance (see SVDiff [1] and AdaLoRA [2] for reference).
  + If it's just an observation, which I think would be more possible (but excuse my bias), then it also makes sense and there is nothing surprising about it: as LoRA has limited size of the parameters, altering the scaling of the transformation might not be enough to fit the downstream data, and hence more rotational transformation are included, which is consistent with another observation in the paper: "The total number of intruder dimensions increases proportionally to the size of the fine-tuning dataset."
- **Chapter 4 is a bit off the topic**: I don't see the connection between the intruder dimension and continual learning in this case study. And it seems quite established that lower rank of LoRA tends to forget more. At the same time, **Chapter 5 is not in-depth enough**: the conjectures given are straightforward and does not inspire new understanding of LoRA and intruder dimension. I would like to see more experiments that specifically target on the assumptions made in Chapter 5.

**References**
- [1] Han, Ligong, et al. "Svdiff: Compact parameter space for diffusion fine-tuning." Proceedings of the IEEE/CVF International Conference on Computer Vision. 2023.
- [2] Zhang, Qingru, et al. "AdaLoRA: Adaptive budget allocation for parameter-efficient fine-tuning." arXiv preprint arXiv:2303.10512 (2023).

**Questions:**

My main questions are included in "Weaknesses".

---

> ### Author Response · Authors · 2024-11-21
> **NsNJ**
>
> We thank you for your review. Per your advice, we conduct our analysis of intruder dimensions on models trained with AdaLoRA.
>
> > According to Def 1, intruder dimensions depend on epsilon, which varies with different number of dimension (footnote 1, page 4).
>
> Epsilon, our cosine similarity threshold, is a hyperparameter in Def. 1. In Fig. 5 we show that our findings about the presence of intruder dimensions in LoRA fine-tuned models and their absence in full fine-tuned models holds for a wide range of \epsilon values.
>
> Please note, the choice of epsilon is in no way related to footnote 1 on page 4. Footnote 1 is intended to build intuition of how you can have a new vector have low cosine similarity with all vectors that span a space. Therefore, our epsilon in Def.1 is unrelated to this footnote and it does not vary with dimension.
>
> > The concept of intruder dimension considers only the rotational effect of a linear mapping, and it holds only the weight increment $\Delta W$ is small compared to the pre-trained weights.
>
> Intruder dimensions do not only consider the rotational effect because we search for intruder dimensions in order of their singular values: this means we are searching for high ranking intruder dimensions, which can be interpreted as learned, highly impactful axes of transformation.
>
> It is true that this measure will only be meaningful if large changes are not made to the initial weight matrix (ex. from initialization to the end of pre-training). However, we show empirically, this is exactly what occurs during fine-tuning: small changes are made to the pre-trained weights. Because of this, our measure an effective lens for studying the changes made to a pre-trained model during fine-tuning.
>
> > Could you please explain why we need to define such a new concept?
>
> Thank you for your thoughtful and detailed question! While your provided formulation is an interesting lens to use, it is a measure that quantifies the change in the matrices but does not pin-point where the change is localized---which in this case is critical. In contrast, our measure is very specific: it allows us to identify and point to the exact singular vector that is different across the pre-trained and fine-tuned weight matrices.
>
> The reason why we need to define a new concept is because we need a way to characterize the new, high ranking singular vectors that are being learned by LoRA during fine-tuning--while the rest of the singular vectors are kept nearly constant.
>
>
> > Are intruder dimensions the reason for worse OOD generalization or the consequence of LoRA?
>
> See general response. We show that intruder dimensions are indeed worse for OOD generalization.
>
>
> Chapter 4:
> > I don’t see the connection between the intruder dimension and continual learning in this case study.
>
> Our continual learning case study measures how much each fine-tuning method forgets previously learned information. We examine how well each method effects the model's performance on the tasks it is previously trained on. In this case study, we see that while LoRA and full fine-tuning initially reach similar accuracies, lower ranks of LoRA forget much more (and therefore have much lower accuracy) as they are trained on new tasks.
>
> > It seems quite established that lower rank of LoRA tends to forget more.
>
> We respectfully disagree--some past work has observed the opposite behavior under some circumstances[1], possibly because it uses fewer parameters and therefore cannot overfit to a downstream dataset as much as full fine-tuning. In our work, we paint a more nuanced picture: while indeed LoRA can forget less in comparison to full fine-tuning, the selection of rank and alpha matters to achieve this.
>
> Chapter 5:
> > the conjectures given are straightforward and does not inspire new understanding of LoRA and intruder dimension.
>
> Chapter 5 consists of empirical observations that justify why intruder dimensions occur and what factors play a role in their occurrence. They are nontrivial and are all backed by experiments or supporting mathematical analysis to further our understanding of intruder dimensions and justify why LoRA has them.
>
> > I would like to see more experiments that specifically target on the assumptions made in chapter 5.
>
> We believe every claim in chapter 5 is...please let us know if there is anything specific that you think is missing.
>
> Citations:
>
> [1] https://arxiv.org/abs/2405.09673

---

> > ### Author Response · Authors · 2024-12-02
> >
> > Since the reviewing window is closing soon, please let us know if there are any more concerns you have with our work that you would like us to address. If you have no more concerns, we ask that you raise your score to reflect this.

---

### Author Response · Authors · 2024-11-20
**General Response (1/2)**

We thank all the reviewers for all their thoughtful feedback. We have added new plots and analysis to the pdf to address the reviewers' concerns.

**Summary of updates to the submission pdf:**
1. Added plot showing that intruder dimensions are responsible for worse OOD performance (Figure 22, Page 23). [NsNJ,UXpu,UCu1]
2. Added LoRA Variants plot (Figure 19, Page 21). [NsNJ,w6KM,UCu1]
3. Updated Figure 7 (added accuracy curve). [UXpu]
4. Added plot showing number of intruder dimensions with respect to number of epochs and learning rate (Figure 20, Page 22). [UCu1]
5. Added plot showing impact of random seed on intruder dimensions (Figure 21, Page 22). [UCu1]

**Intruder Dimensions are responsible for worse OOD performance.**

Several reviewers raised concerns about why intruder dimensions are important or if they are responsible for worse OOD generalization. To provide evidence that intruder dimensions do indeed cause worse OOD performance, we conduct a new experiment in which we scale the top intruder dimension in each matrix by a value, which we call lambda, and examine how this impacts both test acuracy and pre-training loss. Note that $\lambda$=1 will be no change, and $\lambda$=0 will remove the intruder dimension. For our baseline, we instead scale the intruder dimension's neighboring singular vector.
We report this experiment in Figure 22 (Page 23) of the updated submission. In it, we see that if we scale down the intruder dimensions, we get much better OOD performance on the pre-training distribution. For example, in the rank 1 case, we see a **25.9% drop in loss** by zeroing out the intruder dimensions ($\lambda$=0). In the rank 8 case, we see a **26.1% drop in loss** by zeroing out the intruder dimensions. Notably, this is accompanied by a **only a 5.9% drop in performance**. If we instead increase the intruder dimensions ($\lambda$>1), we see that pre-training loss increases significantly while in distribution accuracy does not degrade by much. For example, in the rank 8 case, with lambda=2 we see that **pre-training loss increases by 135%** but **test accuracy only drops by 3.8%**. These results suggest that while the top intruder dimension was responsible for most of the worsened OOD performance in comparison to the pre-trained model, it was also responsible for a portion of the model's learning.
In addition, we study the impact of learning rate on intruder dimensions and pre-training performance and find a strong correlation ($\rho$ = 0.944, p-value $\leq 0.001$) between the appearence of intruder dimensions and worse pre-training performance. This experiment is described in more detail below and can be seen in Figure 20 (Page 22).

**Intruder Dimension Measurements of LoRA Variants.**

As requested by several reviewers, we measure the intruder dimensions of models fine-tuned with the LoRA variants AdaLoRA, LoRA+, PiSSA, & VeRA. We report these values in Figure 19 (Page 21) of the updated submission.

We find several things from this analysis of the intruder dimensions of various methods. We still find that using a higher rank is effective for reducing the number of intruder dimensions after fine-tuning. Importantly, using a low rank still appears to be a very strong indicator of the presence of intruder dimensions. However, certain methods appear to have fewer in comparison to others: AdaLoRA, which reparametrizes the LoRA update as an SVD-like module, appears to have fewer intruder dimensions, suggesting that this methodology of separating the rotational and scaling components may be beneficial for reducing intruder dimensions.

---

> ### Author Response · Authors · 2024-11-21
> **General Response (2/2)**
>
> **Impact of Learning Rate on Intruder Dimensions.**
>
> As recommended study the impact of learning rate on the number of intruder dimensions and measure this across training epochs. These results are reported in Figure 20 (Page 22). When ablating the learning rates that LoRA uses, we observe a clear trend: using a larger learning rate leads to having many more intruder dimensions. This is observed when models, despite their different learning rates, each reach very similar performance. Since we have shown that intruder dimensions are bad for OOD performance, these results suggest a simple prescription: use the smallest learning rate that you can to get to the same accuracy on the target task. (In fact, this experiment further emphasizes the impact of intruder dimensions on pre-training loss: we see a clear relationship between the number of intruder dimensions and worse pre-training loss.)
>
> It is important to emphasize that we are using default or common learning rate values in our experiments. In fact, although we can use 5e-5 as our learning rate and get very few intruder dimensions, most use a larger learning rate: defaults of 1e-4 are generally used [4] but some use even larger learning rate (the original LoRA paper used a learning rate of 5e-4 [1]). Therefore, the phenomenom of intruder dimensions is not a facet of us selecting bad hyperparameters, but rather a *symptom of how LoRA is actually used in the real world today*.
>
> **What About Long-Form Tasks?**
>
> In this work, we aim to be consistent with prior works([1], [2], [3]), which did not benchmark LoRA with code generation or other long form evals but instead used common sense reasoning and sequence classification tasks. We point out that more computational resources are required to fine-tune models for code generation or other long form tasks and is a limiting factor for us. While we do have intruder dimension measurements on models for code generation and other long-form tasks, we do not have pre-training loss experiments for these because it is widely found that LoRA doesn't match the performance of full fine-tuning on tasks like code generation([5]) and long-form text([6]) and therefore any OOD analysis would diverge from our premise of "measuring OOD performance given the same IID performance".
>
>
> **Clarification of Theory**
>
> It is important to point out that intruder dimensions are an empirical finding and are not a requirement of LoRA. In fact, as we have reported in the paper, higher ranks of LoRA do not have as many intruder dimensions or even at all in some cases.
>
> That being said, we have presented several factors that play a role in the introduction of intruder dimensions. We showed that adding the outer product of a random vector to a weight matrix introduces an intruder dimension. This is analogous to the matrix product of BA in LoRA when rank is 1, and implies that intruder dimensions occur because the B and A vectors are uncorrelated to the columns/rows of the weight matrix W_0. Further, we showed empirically that just training the B matrix, while freezing the A matrix with all singular values of 1, eliminates the amplification of the singular values because of the matrix product and therefore reduces the number of high ranking intruder dimensions.
>
> Citations:
>
> [1]: LoRA: https://arxiv.org/abs/2106.09685
>
> [2]: AdaLoRA: https://arxiv.org/abs/2303.10512
>
> [3]: VeRA: https://arxiv.org/abs/2310.11454
>
> [4]: https://huggingface.co/docs/diffusers/training/lora
>
> [5]: LoRA Learns Less: https://arxiv.org/abs/2405.09673
>
> [6]: Tulu 2: https://arxiv.org/abs/2311.10702

---

### Author Response · Authors · 2024-11-23

As the review period is coming to a close, we would greatly appreciate it if you could take a look at our responses and let us know if you have any remaining concerns. If not, we would appreciate it if you could update your score.

---

### Meta-Review · Area_Chair_J8xh · 2024-12-22

**Metareview:**

In this paper, the authors introduce a new concept, the intruder dimension, to investigate the differences between LoRA fine-tuning and full-weight fine-tuning. This concept quantifies the similarity between the column bases of singular-value decomposed matrices of pre-trained and fine-tuned weights. Using the intruder dimension, the authors conduct an empirical analysis by visualizing the performance of various models and their corresponding number of intruder dimensions across multiple datasets. While the concept of intruder dimension is new, the reviewers have concerns about its practical significance and the generalizability of it in different fine-tuning scenarios. I do not think the paper can be accepted in its current form and suggest the authors add more theoretical justifications of intruder dimension and also conduct more experiments with careful designs to show the generalizability of intruder dimension.

**Additional Comments On Reviewer Discussion:**

Reviewer UCu1 has a thorough discussion with the authors, while the other reviewers were not involved in the discussion. I checked the author's response to the other reviewers, and found that the main concern of why the intruder dimension is needed (raised by both NsNj and UCu1) is not well addressed. UCu1 and the authors had a thorough discussion, but the authors failed to convince the reviewer the intruder dimension is really needed to explain the observations in LoRA finetuning.

---

### Decision · Program_Chairs · 2025-01-22

Reject